# Electrochemical oxygen reduction to hydrogen peroxide at practical rates in strong acidic media

Xiao Zhang [1,5✉], Xunhua Zhao [2,5], Peng Zhu [1], Zachary Adler [1], Zhen-Yu Wu [1], Yuanyue Liu [2✉] & Haotian Wang[1,3,4✉]

Electrochemical oxygen reduction to hydrogen peroxide ($H_2O_2$) in acidic media, especially in proton exchange membrane (PEM) electrode assembly reactors, suffers from low selectivity and the lack of low-cost catalysts. Here we present a cation-regulated interfacial engineering approach to promote the $H_2O_2$ selectivity (over 80%) under industrial-relevant generation rates (over 400 mA cm$^{-2}$) in strong acidic media using just carbon black catalyst and a small number of alkali metal cations, representing a 25-fold improvement compared to that without cation additives. Our density functional theory simulation suggests a "shielding effect" of alkali metal cations which squeeze away the catalyst/electrolyte interfacial protons and thus prevent further reduction of generated $H_2O_2$ to water. A double-PEM solid electrolyte reactor was further developed to realize a continuous, selective (~90%) and stable (over 500 hours) generation of $H_2O_2$ via implementing this cation effect for practical applications.

[1] Department of Chemical and Biomolecular Engineering, Rice University, Houston, TX 77005, USA. [2] Texas Materials Institute and Department of Mechanical Engineering, The University of Texas at Austin, Austin, TX 78712, USA. [3] Department of Chemistry, Rice University, Houston, TX 77005, USA. [4] Department of Materials Science and NanoEngineering, Rice University, Houston, TX 77005, USA. [5]These authors contributed equally: Xiao Zhang, Xunhua Zhao. ✉email: xiao1.zhang@polyu.edu.hk; yuanyue.liu@austin.utexas.edu; htwang@rice.edu

Hydrogen peroxide ($H_2O_2$) is ranked as one of the top 10 most energy-intensive chemicals in the chemical manufacturing bandwidth study by the Advanced Manufacturing Office in Department of Energy[1]. It is currently manufactured industrially by the energy- and waste-intensive anthraquinone cycling process[2–4], which consumes a primary current typical energy of ~13,000 Btu/lb (8.1 kWh/kg) without taking into account the $H_2/O_2$ feedstocks[1]. Electrochemical synthesis of $H_2O_2$ via the 2e⁻ oxygen reduction reaction (2e⁻–ORR), where the $O_2$ molecule is electrochemically reduced to $H_2O_2$ via a two-electron (2e⁻) pathway, provides a promising energy-efficient and low-waste alternative[2–7]. Recent efforts have been mostly focused on developing catalysts in alkaline solutions, in which small overpotential and high selectivity of 2e⁻–ORR toward $H_2O_2$ have been comparatively easy to achieve on low-cost materials such as carbon[8–18]. However, in alkaline solutions, $H_2O_2$ is deprotonated ($pK_a > 11$) and easily degraded[19]. Moreover, for practical electrolyzers such as membrane electrode assembly (MEA), catalysts developed in alkaline solutions need to be applied on an anion exchange membrane (AEM), which is typically not as stable as its counterpart of proton exchange membrane (PEM), e.g., Nafion[4], especially operated under air. In addition, with stronger oxidation ability in acid, the acidic $H_2O_2$ solution shows a wider range of applications and greater demand[6,20], which strongly motivates studies in high-performance electrochemical generation of $H_2O_2$ in acidic media.

Till so far, there are only a few known noble metal catalysts, including Pt- and Pd-based catalysts, demonstrated to be selective and stable for 2e⁻–ORR in strong acids[21–24], but their high cost and toxicity of heavy metals (in the case of PtHg alloy[22,23]) could limit their applications in large-scale $H_2O_2$ generation. Some low-cost catalysts such as carbon materials may also show good $H_2O_2$ selectivity in acids within small overpotential and small current density regions (typically less than 10 mA cm⁻²)[25–29], but their $H_2O_2$ selectivity and stability were dramatically dropped when an industrial-relevant current was reached[5,27,30]. In acidic media, carbon catalysts present sluggish ORR kinetics and typically require a large overpotential (>300 mV) to initiate the ORR reaction, and consequently, a large negative cathodic overpotential is required to deliver a high current density[26–29]. While the carbon surface might intrinsically prefer a 2e⁻–ORR pathway due to their relatively weak binding with oxygen intermediates as demonstrated by several previous studies[26,27,30,31], such negative overpotential could further push the ORR reaction all the way down to $H_2O$ with significantly decreased $H_2O_2$ selectivity and production rate especially in acids. This is because under negative potentials in acids, the catalyst surface accumulates concentrated protons that are prone to further reduce the locally generated $H_2O_2$ molecules to $H_2O$ ($H_2O_2 + 2e^- + 2H^+ = 2H_2O$). Therefore, diluting the local proton concentration and minimizing the electrochemical dissociation of as-produced $H_2O_2$ to $H_2O$ could be one promising strategy for resolving this $H_2O_2$ selectivity-activity dilemma, and delivering industrial-relevant $H_2O_2$ production rate in acidic solution while maintaining good $H_2O_2$ selectivity[32,33] (Fig. 1).

## Results

### The alkali metal cation effect on acidic $H_2O_2$ generation in flow cell.

We first employed a standard three-electrode flow cell reactor to investigate the cation effect towards $H_2O_2$ generation in acid (Fig. S1), allowing to evaluate our hypothesis in a more practical environment and produce $H_2O_2$ under higher current densities compared to the traditional RRDE setup (Supplementary Note 1, Fig. 2a). The commercially available carbon black catalyst (BP2000) with a high surface area was used as a model ORR catalyst in this study (Fig. S2). Its intrinsic $H_2O_2$ activity and

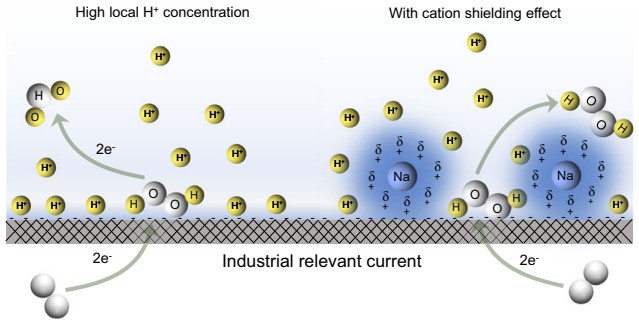

High local H⁺ concentration    With cation shielding effect

2e⁻    Industrial relevant current    2e⁻

**Fig. 1 Schematic illustration of 2e⁻–ORR toward $H_2O_2$ in acid with/without Na⁺ under industrial-relevant current.** Here we report a cation-regulated catalyst/electrolyte interface to promote electrochemical $O_2$ reduction to $H_2O_2$ in acids with high-selectivity and industrial-relevant production rates. By adding only a small amount of alkali metal ions into the acidic electrolyte, which barely affects the solution's pH, we demonstrated a dramatic improvement in $H_2O_2$ selectivity and activity especially under large ORR current densities across different catalysts. Our molecular dynamic simulations suggest that the solvated alkali metal cations, compared to concentrated protons in acids, could preferentially be attracted to the catalyst/electrolyte interface and squeeze out local protons during the reaction, suppressing the further reduction of as-generated $H_2O_2$ to $H_2O$ (Fig. 1). Using commercial carbon black catalysts with 10 mM $Na_2SO_4$ as an additive, the $H_2O_2$ Faradaic efficiency (FE) can reach over 80% under a significant current of 400 mA cm⁻² in 0.1 M $H_2SO_4$, representing a 25-fold improvement compared to the case without Na⁺ additive where negligible $H_2O_2$ was produced (<5% FE). Based on this cation promotion concept, a double-PEM-based solid electrolyte (SE) reactor was developed for a continuous generation of $H_2O_2$ with high FE (~90%) and good stability (over 500 h) for practical applications in the future.

selectivity were first evaluated in 0.1 M $H_2SO_4$ electrolyte (pH = 0.96) (Fig. 2b, c). We observed that while the $H_2O_2$ FE of carbon black catalyst in acid remained relatively good (~70%) under small current regions, it started to decrease dramatically once the current density is over 100 mA cm⁻² with very negative applied potentials (Fig. 2c). Under 200 mA cm⁻², the carbon black catalyst can only deliver a 35% $H_2O_2$ FE and the majority of electrons were directed towards $H_2O$ instead (Fig. 2c). The decreased FE is as expected, because under such a negative potential of −0.89 V versus reversible hydrogen electrode (vs. RHE) needed to drive this high current, even if the catalyst prefers to reduce $O_2$ to $H_2O_2$ in its first place, those generated $H_2O_2$ at the electrode surface could be easily further reduced to $H_2O$ coupling two electrons and two local protons. Please be noted here that, under each current density, its corresponding $H_2O_2$ FE was measured within 8 min of operation. With a longer time of electrolysis, the $H_2O_2$ FE could be further dropped (as shown in the stability test in Fig. 3g). After introducing a trace amount of Na⁺ additive (5 mM $Na_2SO_4$) into the acidic electrolyte, while the ORR activity did not show that much difference (Fig. 2b), the $H_2O_2$ FE was significantly improved especially under high current densities. As shown in Fig. 2c, in the potential range of the ORR onset, the impact of Na⁺ (5 mM $Na_2SO_4$) toward $H_2O_2$ is negligible; while in the range of large overpotentials at high current densities, the small amount of Na⁺ helped the carbon black catalyst to hold a high $H_2O_2$ FE plateau of over 80% until 200 mA cm⁻², suggesting a more than doubled FE compared to that in pure acid (Fig. 2c).

The measured trends of $H_2O_2$ FE show that the promotion effect of Na⁺ is more pronounced at high current density toward $H_2O_2$ formation. We understand that Na⁺ and protons are both positively charged ions and will be attracted towards the ORR

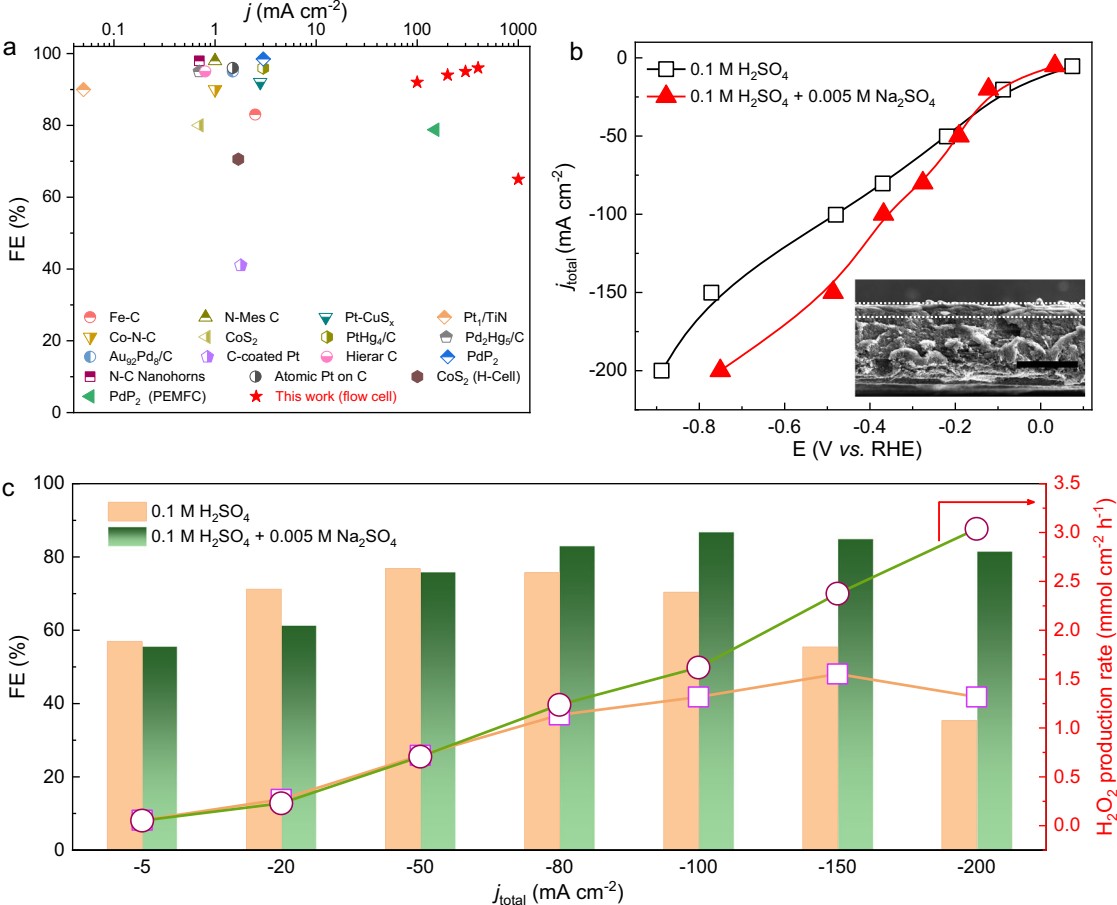

**Fig. 2 Na$^+$ effects toward acidic H$_2$O$_2$ generation through 2e$^-$–ORR. a** Reported current densities and Faradaic efficiencies (FEs) for 2e$^-$–ORR toward H$_2$O$_2$ in acid in literatures (listed in Table S1). **b** Effect of 0.005 M Na$_2$SO$_4$ on the total activity over carbon catalysts BP2000 in a flow cell. The cell voltages were 85% iR compensated. The inset is the cross-section SEM image of the electrode. The scale bar is 200 μm. **c** The comparison of H$_2$O$_2$ FE and production rate over carbon catalyst BP2000 with/without 0.005 M Na$_2$SO$_4$ in a flow cell. The Na$^+$ serves as a promoter for the production of H$_2$O$_2$ through 2e$^-$–ORR. The H$_2$O$_2$ FE and production rate are both increased, especially at the high current range.

catalyst surface to form the electrochemical double layer under negative potentials[34]. We would assume that, Na$^+$ ions could be more competitive than protons to be aligned along the electrochemical double layer, which dramatically reduces the local proton concentration and thus protects the generated H$_2$O$_2$ from further reductions coupling protons and electrons (discussed in the following sections in detail). To further amplify the cation effect and drive the O$_2$-to-H$_2$O$_2$ production at even higher current densities in acid (>200 mA cm$^{-2}$), we gradually increased the Na$^+$ cation concentration in the electrolyte. As shown in Fig. 3a, the overall current density, as well as the H$_2$O$_2$ FE gradually increases with increased Na$^+$ concentrations. In general, a higher concentration of Na$^+$ can maintain larger 2e$^-$–ORR currents without sacrificing the H$_2$O$_2$ selectivity. With only 0.01 M Na$_2$SO$_4$ in 0.1 M H$_2$SO$_4$, the FE of H$_2$O$_2$ can reach 83% at 400 mA cm$^{-2}$ (Fig. 3b), representing a 25-fold improvement compared to that in pure acids without Na$^+$ (H$_2$O$_2$ FE only 3.3%). Along with the increased H$_2$O$_2$ FE, the improvement of the H$_2$O$_2$ production rate was also obvious at high current densities (Fig. 3c). For example, the production rate of 6.21 mmol cm$^{-2}$ h$^{-1}$ H$_2$O$_2$ (partial current of 332 mA cm$^{-2}$) was achieved under the current density of 400 mA cm$^{-2}$, much higher than that in pure H$_2$SO$_4$ acid (only 0.245 mmol cm$^{-2}$ h$^{-1}$). The production rate can be further enhanced by providing more Na$^+$ cations (the production rate of 6.52 mmol cm$^{-2}$ h$^{-1}$ was achieved at 400 mA cm$^{-2}$ and the

maximum FE can be up to 94% at 150 mA cm$^{-2}$ by using 0.05 M Na$_2$SO$_4$ as additive). Further increasing the concentration of Na$^+$ cations could continually push up the H$_2$O$_2$ FE to higher values at high current densities (Fig. S3). It is important to note here that the electrolyte pH did not show obvious change and stayed around pH 1 after these Na$_2$SO$_4$ additives, ranging from 0.96 (0.1 M H$_2$SO$_4$), 0.96 (0.1 M H$_2$SO$_4$ + 0.005 M Na$_2$SO$_4$), 0.98 (0.1 M H$_2$SO$_4$ + 0.01 M Na$_2$SO$_4$) to 1.04 (0.1 M H$_2$SO$_4$ + 0.05 M Na$_2$SO$_4$). To fully exclude the pH effect of the bulk solution (even though the change is quite small), the pH value of 0.1 M H$_2$SO$_4$ + 0.1 M Na$_2$SO$_4$ (pH = 1.13) was adjusted to be the same as that of 0.1 M H$_2$SO$_4$ (pH = 0.96) by adding more sulfuric acid. As shown in Fig. S4 and S5, the H$_2$O$_2$ FE in both electrolytes with Na$^+$ additives (before and after pH tuning) showed very similar trend in all current ranges, indicating the Na$^+$ cations dominate the H$_2$O$_2$ generation process and the small pH differences of electrolytes have negligible influence on the production rate or H$_2$O$_2$ FE. With 0.1 M Na$_2$SO$_4$ as an additive in 0.2 M H$_2$SO$_4$ solution (pH = 0.76), we were able to produce H$_2$O$_2$ at the current density of 1 A cm$^{-2}$ with a FE of more than 65% (Fig. S6). The H$_2$O$_2$ partial currents of up to 650 mA cm$^{-2}$ were achieved, and high FEs were maintained, better than the highest O$_2$-to-H$_2$O$_2$ conversion rates reported. At even lower pH electrolyte, i.e., 1 M H$_2$SO$_4$ solution (PH~0), similar trends were also observed (Fig. S7), indicating the general phenomenon of cation promotion effect toward H$_2$O$_2$ production through ORR.

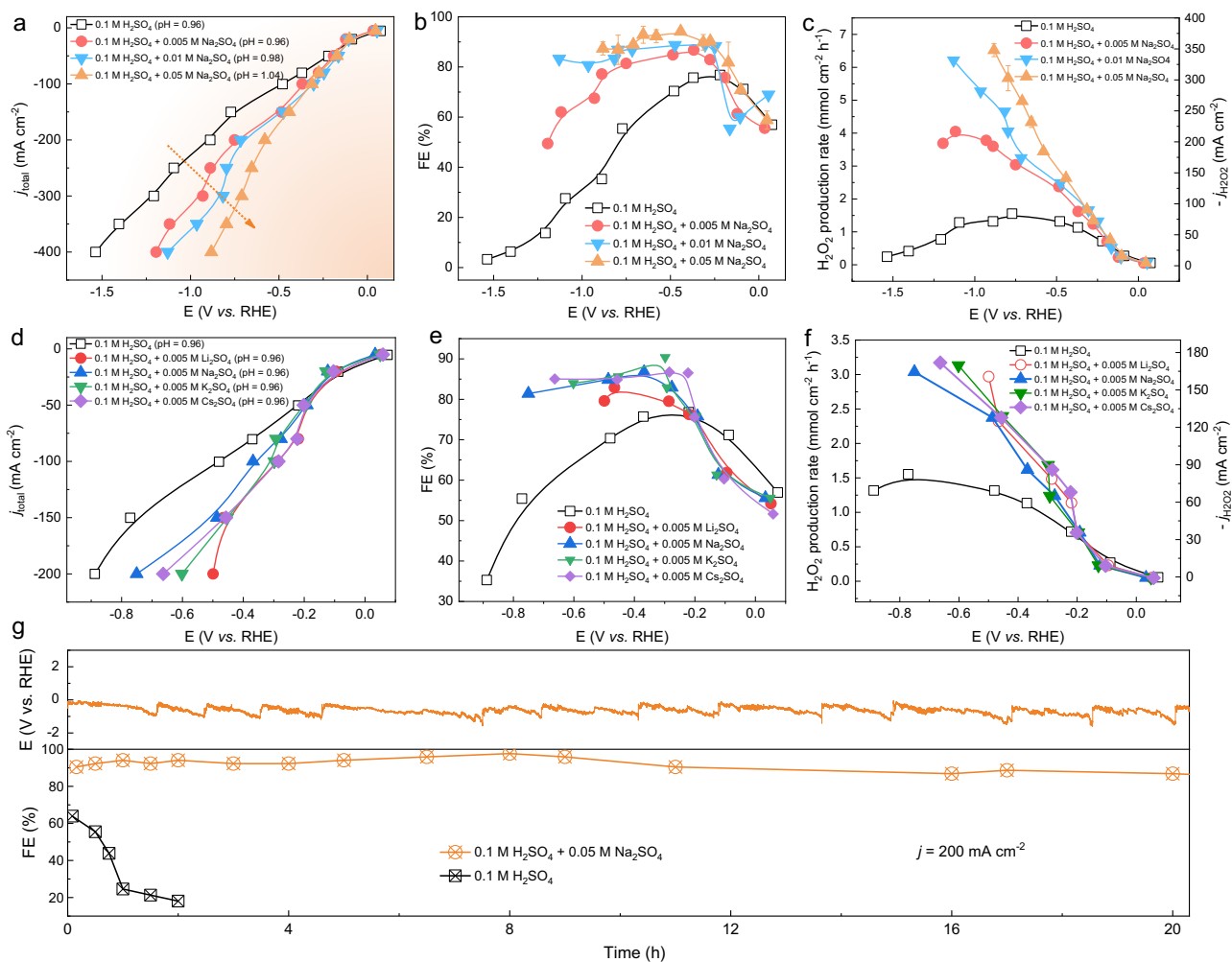

**Fig. 3 The effect of alkali metal cation concentration and species on electrosynthesis of H$_2$O$_2$ through 2e$^-$-ORR. a** The I-V curve of ORR with different concentrations of cations (from 0 to 0.05 M Na$_2$SO$_4$) in a flow cell. **b** The corresponding FEs and (**c**) production rates and partial current of H$_2$O$_2$ products under different cell voltages. With the increase of cation concentration, the H$_2$O$_2$ FE and production rate continuously increase at high current densities. The error bars represent two independent tests. **d** The I-V curve of ORR in 0.1 M H$_2$SO$_4$ containing different types of cations (0.005 M X$_2$SO$_4$, X = Li$^+$, Na$^+$, K$^+$, Cs$^+$) in a flow electrolyte cell. **e** The corresponding FEs and (**f**) production rates and partial current of H$_2$O$_2$ products under different cell voltages. **g** Chronopotentiometry stability comparison of carbon catalysts BP2000 with/without cations by holding 200 mA·cm$^{-2}$ current density for 20 h in a flow cell for continuous electrolysis. The geometric area of the flow field at the cathode in the flow cell is 1 cm$^2$.

The low threshold of the alkali metal cation concentration towards promoting H$_2$O$_2$ generation puts forward new requirements for the purity of the electrolyte during ORR to H$_2$O$_2$ tests in acids. For the traditional electrolytic ORR process, the Na$_2$SO$_4$ is widely used as the anolyte to balance the electrochemical reaction. However, even far away from the cathode side and separated by ion exchange membranes, we found that the Na$^+$ in the anolyte can still penetrate the PEM and move to the cathode chamber, and thus significantly improve the H$_2$O$_2$ FE of ORR at the cathode (Fig. S8). Therefore, it is highly recommended to perform the ORR reaction using the same acidic electrolyte to avoid any cross-over cation contaminations which could significantly improve H$_2$O$_2$ performance in acids.

The promotion effect is not only limited to the Na$^+$. The H$_2$O$_2$ production rate can also be enhanced by using other alkali metal ions. Figure 3d shows the I-V curves for four different alkali metal cations in each sulfuric acid electrolyte with a concentration of 0.01 M (0.005 M X$_2$SO$_4$, X = Li, Na, K, Cs). As compared with the pure H$_2$SO$_4$ electrolyte, while the ORR activities were slightly improved, significant improvements in H$_2$O$_2$ FE were observed for all the cations (Fig. 3d–f). The FE and production rates of

H$_2$O$_2$ are relatively unaffected by the size of the alkali metal cations in the electrolyte. As differences in total current density exist for electrolytes containing different cations, the H$_2$O$_2$ production rate provides a better representation of trends in product formation rates than FEs. As shown in Fig. 3f, with 0.005 M X$_2$SO$_4$ as the additive, all the alkali metal cations are able to drive the O$_2$-to-H$_2$O$_2$ reaction efficiently with high production rates, and the differences induced by different cations are relatively marginal. Nevertheless, we find that the promotion effect is only limited to IA alkali metal cations (such as Li$^+$, Na$^+$, K$^+$, Cs$^+$), while the other cations (including the IIA cations such as Mg$^{2+}$, Ca$^{2+}$ and IIIA cations such as Al$^{3+}$) decrease the H$_2$O$_2$ FE dramatically. This might due to the local environment change from acid to alkaline induced by the cation additives during ORR (will discuss the details in the simulation part). The alkaline local environment could induce the formation of solid metal hydroxide on the catalyst surface and block the ORR reaction, decreasing the H$_2$O$_2$ FE and production rate (Fig. S9).

It was also exciting to find out that the cation additives not only promote the H$_2$O$_2$ selectivity but also improve the long-term operation stability, which is another important target for practical

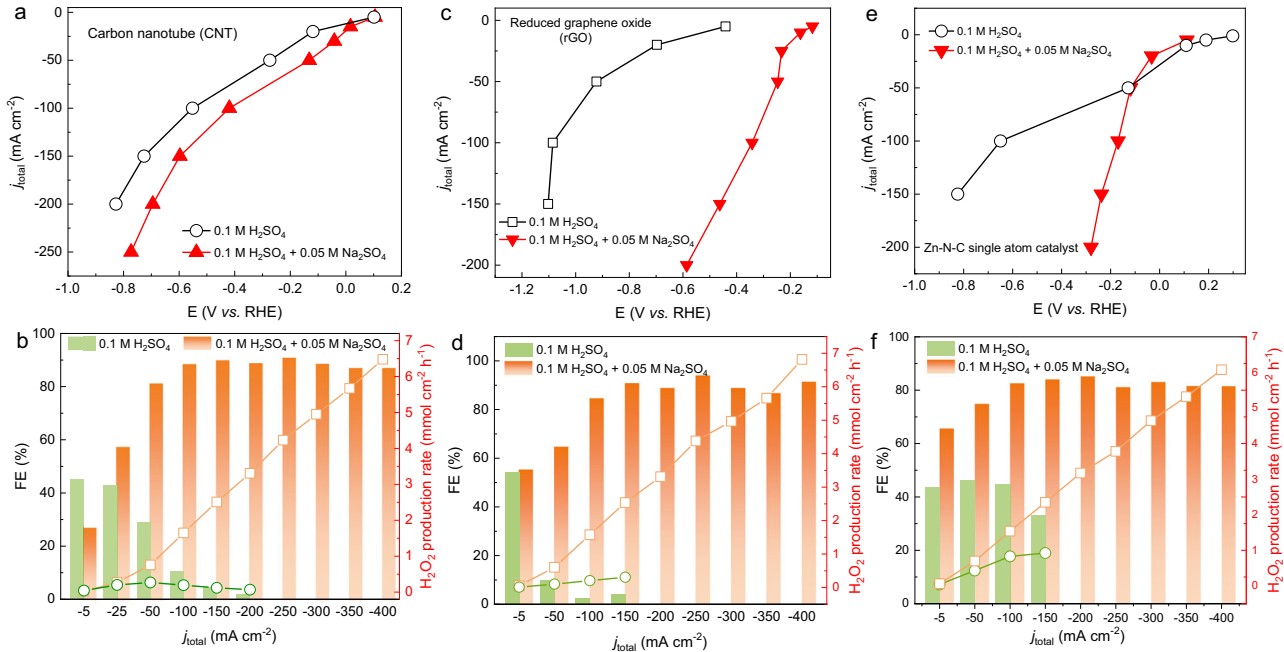

**Fig. 4 The Na⁺ effect towards electrochemical production of $H_2O_2$ on diverse carbon-based electrocatalysts.** The I-V curve of ORR in 0.1 M $H_2SO_4$ or 0.1 $H_2SO_4$ + 0.05 M $Na_2SO_4$ electrolytes using (**a**) CNT, **c** rGO, **e** Zn-N-C single-atom catalysts. The corresponding FEs and production rates of $H_2O_2$ in 0.1 M $H_2SO_4$ or 0.1 $H_2SO_4$ + 0.05 M $Na_2SO_4$ electrolytes using (**b**) CNT, **d** rGO, **f** Zn-N-C single-atom catalysts.

production of $H_2O_2$. Figure 3g shows the comparison of $H_2O_2$ FE at 200 mA/cm² ORR current as a function of operation time in the flow cell. In pure acidic electrolyte, the $H_2O_2$ FE rapidly dropped to less than 10% within 2 h. As a sharp contrast, with the presence of Na⁺ cations, the potential and $H_2O_2$ FE showed negligible changes for over 20 h. We suppose the improved activity and stability are induced by the alkalization of the local environment induced by the cation additives during ORR. During the ORR process, the solvated alkali metal cations, compared to concentrated protons in acids, could preferentially be attracted to the catalyst/electrolyte interface, which gives rise to a local alkaline environment (will discuss the details in the simulation part). At alkaline conditions, the carbon-based catalyst typically shows higher activity and better stability compared to acidic conditions[8–10]. We then found this cation promotion effect has broad applicability to different catalysts and acidic electrolytes. With a small amount of $Na_2SO_4$ as the additive in 0.1 M $H_2SO_4$ (Fig. 4), high $H_2O_2$ FE under high production rates can be achieved on various carbon catalysts including carbon nanotube (CNT), reduced graphene oxide (rGO), XC-72, and the carbon-based single-atom catalysts (e.g., Zn-N-C). This promotion effect is independent of the catalyst structures, surface morphology or surface area (Figs. S10–S12). In addition, this cation impact on 2e⁻–ORR also applies to other acidic electrolytes such as $HClO_4$, and the Na⁺ sources can also come from other salts such as $NaHSO_4$ (Figs. S13, S14).

**Mechanistic understanding of the cation effect on $H_2O_2$ selectivity.** To further explore our aforementioned hypothesis and understand the cation effects on 2e⁻–ORR selectivity, we carried out constant-potential *ab*-initio molecular dynamics (AIMD) simulations[35] based on DFT. In the simulations, we use a typical potential $V_{RHE}$ (potential relative to RHE) = −1 V at which the cation effect is most prominent. The hydrogen evolution reaction was not considered because only a trace amount of $H_2$ byproduct (from $H_2$ evolution at large overpotentials) was detected from the cathode side (Fig. S15). We consider the

acid condition and use pH = 0 in our simulations. Since alkali metal cations are not likely to directly participate in ORR, we explore two aspects of the cation effects: (1) how alkali metal cations affect the distribution of the protons, (2) how the redistribution of protons will influence the selectivity of $O_2$ reduction to $H_2O_2$. For the first question, we adopted a model with relatively thick water layers (equivalent to 6 ice layers) on (6 × 6) periodic graphene (Fig. S16), and then put a certain number of cations/protons to represent the electrode environment under the lower potential like $V_{RHE}$ = −1 V. For the second question, we use a thinner model (~4 ice layers) with single-vacancy to represent the reaction site, and then use slow-growth method[36] to evaluate the reaction barrier under different conditions). The details of the simulations can be found in the experiment section in supporting information (SI).

As shown in Fig. 5a, b, both Na⁺ cations drift towards the surface in molecular dynamics. This is not unexpected considering that under the low potential $V_{RHE}$ = − 1 V, the surface is charged by ~3e⁻. Such a fast drift may have two consequences: firstly, the local concentration of Na⁺ can be much higher than that in the bulk; secondly, as a charge carrier that compensates the net charge of the substrate, Na⁺ may compete with protons which is the major charge carrier in the acid electrolyte when Na⁺ cations are not added. Indeed, as shown in Fig. S17, when Na⁺ cations and protons co-exist near the interface, the cations compete with the protons by repelling the proton away from the surface. After 3 picoseconds of AIMD, both cations stay ~4.5 Å from the surface, while the protons end at ~8 Å from the surface. These results clearly suggest that the cations, which can be enriched by the attraction of the negatively charged surface, can strongly repel the local protons, and thus dramatically reduce the local proton concentrations.

Then we consider how the selectivity of the 2e⁻–ORR is affected by the presence of Na⁺ cations and the reduced local proton concentrations. We understand that carbon catalysts typically bind oxygen intermediates weakly and thus present an intrinsic selectivity towards $H_2O_2$, which can be seen from our above experimental results (high $H_2O_2$ selectivity under low currents in Fig. 2c) as well as

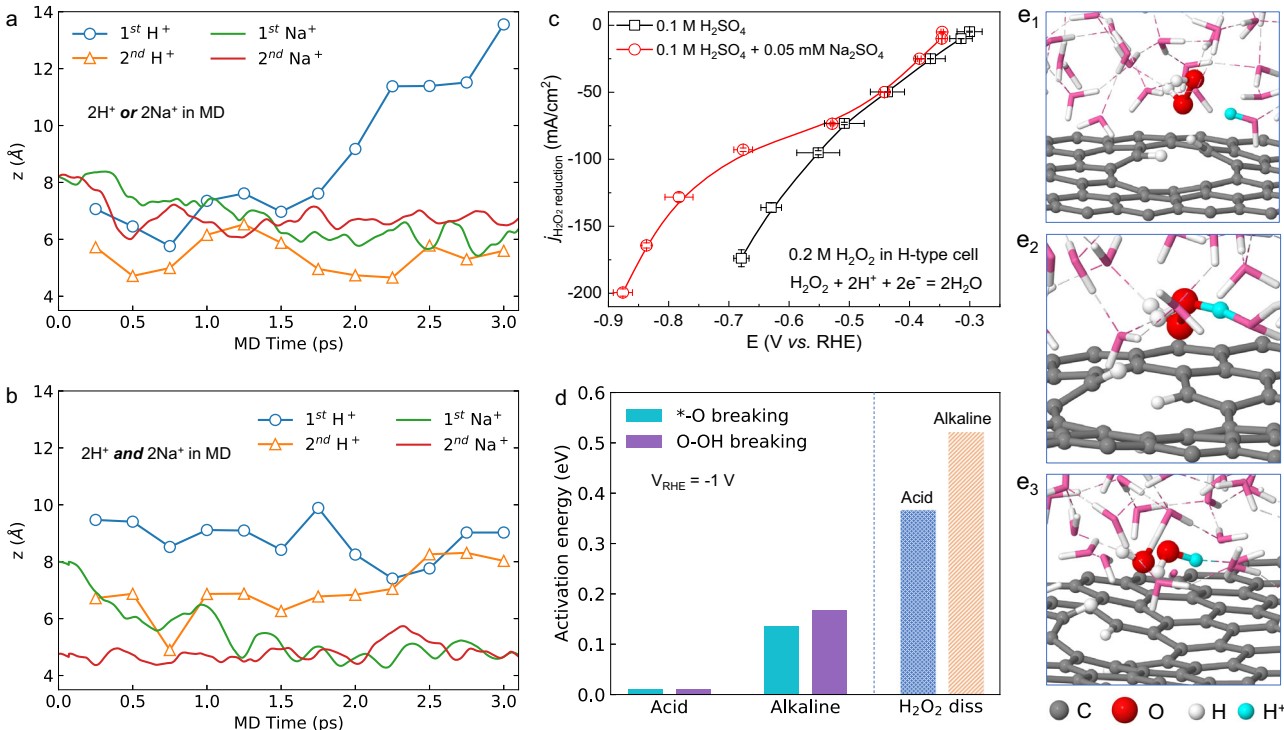

**Fig. 5 Electrochemical $H_2O_2$ dissociation and simulation of the $Na^+$ effect at the local environment of catalyst surface. a**, **b** Evolution of the position (surface normal direction "z") of $Na^+$ ions (and/or) $H^+$ after positioning them in the water layers; **c** the partial current of electrochemical $H_2O_2$ dissociation as a function of the voltage by using 0.1 M $H_2SO_4$ or 0.1 M $H_2SO_4$ + 0.05 M $Na_2SO_4$ electrolyte in H-type cell. The measured potentials were manually 100% compensated. The error bars represent two independent tests. **d** activation energies calculated through constant potential molecular dynamics (see SI for more details). **$e_1$–$e_3$**, Initial state, transition state and final state of the $H_2O_2$ decomposition in neutral solution. C: grey; O: red; H: white; Proton: cyan.

previous reports[9,11]. The catalyst's $H_2O_2$ selectivity starts to drop under significant overpotentials to deliver large currents in pure acids, where two possible reaction mechanisms could play a role in guiding the reaction towards the $4e^-$ pathway of $H_2O$. One possibility is that, while the catalyst still produced $H_2O_2$ selectively in the first place, those locally generated $H_2O_2$ under such negative potentials could be further reduced to $H_2O$ coupling electrons and protons ($H_2O_2 + 2e^- + 2H^+ = 2H_2O$), resulting in a low apparent $H_2O_2$ selectivity. In this case, as the $Na^+$ cation additives can effectively screen away local protons, the further dissociation of generated $H_2O_2$ can be depressed, resulting in better $H_2O_2$ selectivity. This "cation protection" of as-formed $H_2O_2$ can be validated by performing the electrochemical reduction of $H_2O_2$ in both pure and $Na^+$-containing acidic electrolytes (Figs. S19–21, Fig. 5c). As shown in Fig. 5c, the $H_2O_2$ reduction activity was greatly suppressed, especially under large overpotentials, when $Na^+$ was added, suggesting that the introduction of cations can greatly inhibit the $H_2O_2$ dissociation to $H_2O$ under reductive potential environments. Please be noted here that the only possible side reaction, the hydrogen evolution reaction, was also taken into consideration when measuring the $H_2O_2$ reduction currents (Fig. S15). This suppression effect can be further validated from our simulation results. Clearly, as shown in Fig. 5d, the absence of protons increases the $H_2O_2$ decomposition barrier to 0.519 eV, 0.156 eV higher than that when a proton is present. The initial state, transition state and final state of $H_2O_2$ decomposition are shown in Fig. 5e and S17. As the O-O bond elongates, the proton attaches to one of the O in $H_2O_2$ and forms a $HO-OH_2$ complex at transition state and eventually forms $OH^-$ and $H_2O$. This process clearly shows how the proton promotes $H_2O_2$ decomposition, explaining the reason why cations could prevent the further reduction of as-synthesized $H_2O_2$ by screening out local protons.

Another possible factor for improved $H_2O_2$ selectivity is that the cations could suppress the dissociation process of peroxide intermediate (*OOH) during ORR due to local proton depletion. To explore this hypothesis, we further used a slow-growth approach based on AIMD to evaluate the reaction barriers of both $2e^-$ and $4e^-$ paths under different proton concentration conditions. As displayed in Fig. 5d, when a proton is present, both the $2e^-$ path and $4e^-$ paths have an extremely low barrier and take place spontaneously at 300 K. In contrast, the absence of protons increases the *-O breaking ($2e^-$ path) barrier to 0.136 eV, in comparison to 0.167 eV for O-OH breaking ($4e^-$ path), suggesting $2e^-$ pathway being favorable by $\exp((0.167-0.136)k_BT) = 3.36$ times than the $4e^-$ pathway and thus enhanced $H_2O_2$ selectivity. Therefore, the local absence of protons, a result of cation accumulation near the surface, can strongly enhance the $H_2O_2$ selectivity in acids.

We also evaluate the effect of $Mg^{2+}$ on the $2e^-$–ORR by replacing two $Na^+$ by one $Mg^{2+}$, and running the AIMD simulations similar to the case of $Na^+$ (Fig. 5b). As shown in Fig. S27, $Mg^{2+}$ does not show the same effect as $Na^+$. This is likely due to two reasons: (1) $Mg^{2+}$ is more efficiently screened as the $Mg^{2+}$ bonds stronger with O of $H_2O$ than the case of $Na^+$, which is evidenced by the significantly shorter distance between Mg and O (averaged distance $d_{Mg-O} = 2.05$ Å vs. $d_{Na-O} = 2.55$ Å); (2) the mole concentration of $Mg^{2+}$ is only half of that of $Na^+$ in the electrical double layer and the electrostatic repelling decays in the form of $1/r$, so there is more "screened" space for $H^+$ in the electrical double layer.

**Practical generation of $H_2O_2$ using cation exchange membrane solid electrolyte reactor.** Obtaining good $H_2O_2$ selectivity and activity in acidic ORR is a prerequisite for practical implementations of membrane electrode assembly (MEA) reactor using

reliable and well-established PEM such as Nafion (sulfonated tetrafluoroethylene based fluoropolymer-copolymer membrane). However, till so far, only noble metal catalysts such as PtHg, PdHg or $PtP_2$ nanocrystals could deliver reasonable $H_2O_2$ selectivity and stability in proton exchange MEA device[22,23,37]. This observed notable promotion effect of cations on low-cost and non-toxic carbon catalysts in acidic $H_2O_2$ generation therefore provides us with a great opportunity to deliver practical $H_2O_2$ activity, selectivity, and stability. Our basic assumption is that, as the alkali metal cations can move across the PEM, they may help to regulate the local environment of the catalyst/membrane interface for better $H_2O_2$ activity and selectivity. To explore how to successfully make use of this cation tuning effect, we first evaluated its applicability in a traditional PEM-MEA cell configuration (Figs. S22–S24). First, it is well within our expectation that the traditional PEM-MEA cell using commercial carbon black catalyst with 0.1 M $H_2SO_4$ as the anolyte, presented negligible $H_2O_2$ selectivity, due to the high proton flux at the catalyst/membrane interface (Fig. S22). However, we found out that even with the addition of cations in acids (Fig. S23), or directly using $Na_2SO_4$ solution as the anolyte (Fig. S24), no obvious improvements were observed. This is because on the anode side, a significant number of protons will be generated locally at the catalyst/membrane interface during the oxygen evolution reaction and then immediately transported across the membrane to the cathode side, suppressing the possibility of $Na^+$ ion transportation from the bulk anolyte towards the cathode to regulate the interfacial environment of improved $2e^-$–ORR. The low $H_2O_2$ selectivity in PEM-MEA drove us to design a new cell configuration to employ this cation effect for continuous production of $H_2O_2$ in a practical way.

Instead of the traditional PEM-MEA cell design, here we developed a SE reactor with three chambers separated by two PEMs to fully implement this cation effect for high-performance $H_2O_2$ generation (Fig. 6a and Fig. S25). Specifically, the cathode (carbon black) and anode ($IrO_2$) of our device are catalyst-coated GDL electrodes, which were separated by a thin SE layer sandwiched by two identical PEMs (Nafion-117). The cathode side was continuously supplied with a mixture of $O_2$ stream and water flow for $2e^-$–ORR, while the anode side was circulated with $H_2O$ for water oxidation. In the middle chamber, a SE layer consisting of porous polymer ion conductors was used to minimize the $iR$-drop between cathode and anode[38,39]. A dilute cation solution flows through this SE layer to introduce the cation effects on the cathode side $2e^-$–ORR. Please be noted here that without this SE layer, the cell voltage was significantly increased due to the increased cell resistance between the cathode and anode (Fig. S26). Under a negative reduction potential, the cations in the middle SE chamber are driven by the electrical field to penetrate the PEM toward the cathode surface and thus regulate the local environment at the catalyst/PEM interface to promote $H_2O_2$ generation (Supplementary Note 2 and Fig. S27). The $H_2O_2$ molecules formed at the cathode side are then efficiently brought out via the oxygen and DI water flow stream. Meanwhile, protons generated from water oxidation at the anode penetrate the righthand side PEM and move into the middle chamber to compensate for the charge.

The I–V curve of our 4-cm$^2$ three-chamber PEM SE cell flowing 0.03 M $Na_2SO_4$ solution in the middle layer is plotted in Fig. 6b. Of note that the concentration of $Na_2SO_4$ can be varied to higher values. Our target is to realize the high production rate of $H_2O_2$ while minimizing cations' consumption for practical demonstrations. Therefore, a 0.03 M $Na_2SO_4$ solution is adopted to regulate the interfacial environment of improved $2e^-$–ORR in our SE cell. A mixture of $O_2$ gas flow (180 sccm) and DI water flow (1.8 mL·min$^{-1}$) was supplied to the cathode. The flow of DI

water is to efficiently bring out the generated $H_2O_2$. By ramping up the overall current density, the cell voltage of the SE reactor gradually increased. The $H_2O_2$ FE remained over 85% across the entire cell voltage range, with a maximum of 96% at 5 and 20 mA cm$^{-2}$ (Fig. 6b), much higher than the traditional MEA cell configuration. In comparison, the reactor without alkali metal cations requires a higher potential to deliver and shows much lower $H_2O_2$ FE, practically at high current densities (Fig. S28). The crosse-over $Na^+$ from the middle SE layer to the cathode plays a key role in determining the $H_2O_2$ FE (Fig. S27).

The electrolysis stability is always one of the most important but challenging parts in practical applications. Benefiting from the stable material properties in carbon black and PEM as well as reliable cation effects, our SE cell with the double-PEM configuration presents excellent long-term stability in producing $H_2O_2$. The SE reactor stability was evaluated by holding a 50 mA·cm$^{-2}$ cell current density (200 mA total current). As shown in Fig. 6c and d, when supplying dilute $Na^+$ ions in the SE layer, a continuous generation of $H_2O_2$ solution on the cathode side can be stably operated for over 500 h with no degradations in product FE (~90%). As a sharp contrast, in the absence of cations, the $H_2O_2$ FE started with less than 60% and rapidly dropped to less than 10% in 6 h, which can even be recovered to ~90% when cations were later introduced (Fig. S29), clearly suggesting this prominent cation effect. The observation indicates that the carbon catalyst still works well, and this is not the reason for FE degradation. We suppose the FE degradation is because of the accumulation of local $H^+$ to high concentrations with the extension of reaction time, which accelerates the further reduction of $H_2O_2$ to $H_2O$ and decrease the $H_2O_2$ selectivity.

A continuous supply of cation solutions in the SE layer could limit the device's real applications due to the following two reasons. First, it could result in a significant consumption of cations as most of them are flowing out of the SE layer and only part of them crossed to the cathode chamber. Second, the generated $H_2O_2$ solution in the cathode would be slightly alkaline due to the $Na^+$ ion crossover from the SE layer (resulting in slight acidity in the SE layer downstream flow). To further explore the high potential of our SE cell for practical uses, we operated the cell by circulating the cation solution from the SE layer into the cathode side, and then back to the SE layer for $H_2O_2$ accumulation. The outlet solution in the middle chamber was mixed with oxygen flow and supplied to the cathode to produce $H_2O_2$. Then the as-produced solution containing $H_2O_2$ and cations was cycled back into the middle chamber (Fig. 6e). By doing so we could continuously reuse the cations by circulating them back to the SE layer for a closed system without the need for a continuous cation stream supply. Also, the excessive $OH^-$ groups generated from the cathode would be neutralized by exactly the same number of excessive protons in the SE layer. Our target is to accumulate the $H_2O_2$ concentration to ~5000 ppm in a 250 mL solution (containing only 60 mM of $Na^+$) for each operation cycle via maintaining a 50 mA cm$^{-2}$ cell current. As shown in Fig. 6f, the $H_2O_2$ concentration continuously increased to ~5000 ppm in about 13 h. The cell can be operated for more than 200 h with negligible degradations. During this stability test, a total of 3.7 L 5000 ppm $H_2O_2$ solution was obtained. We also observed that, in each operation cycle the $H_2O_2$ FE slightly decreases with increased $H_2O_2$ concentration, which may be due to the $H_2O_2$ self-decomposition, further reductions on the cathode, and/or the crossover oxidation on the anode side. For each operation cycle, the $H_2O_2$ concentration can reach up to ~0.15 M, which is 5 times of the $Na_2SO_4$ additive in the final product. A higher concentration of $H_2O_2$ solutions can also be produced by extending the operation time while maintaining the current and FE. As a result, a high concentration of 20,000 ppm

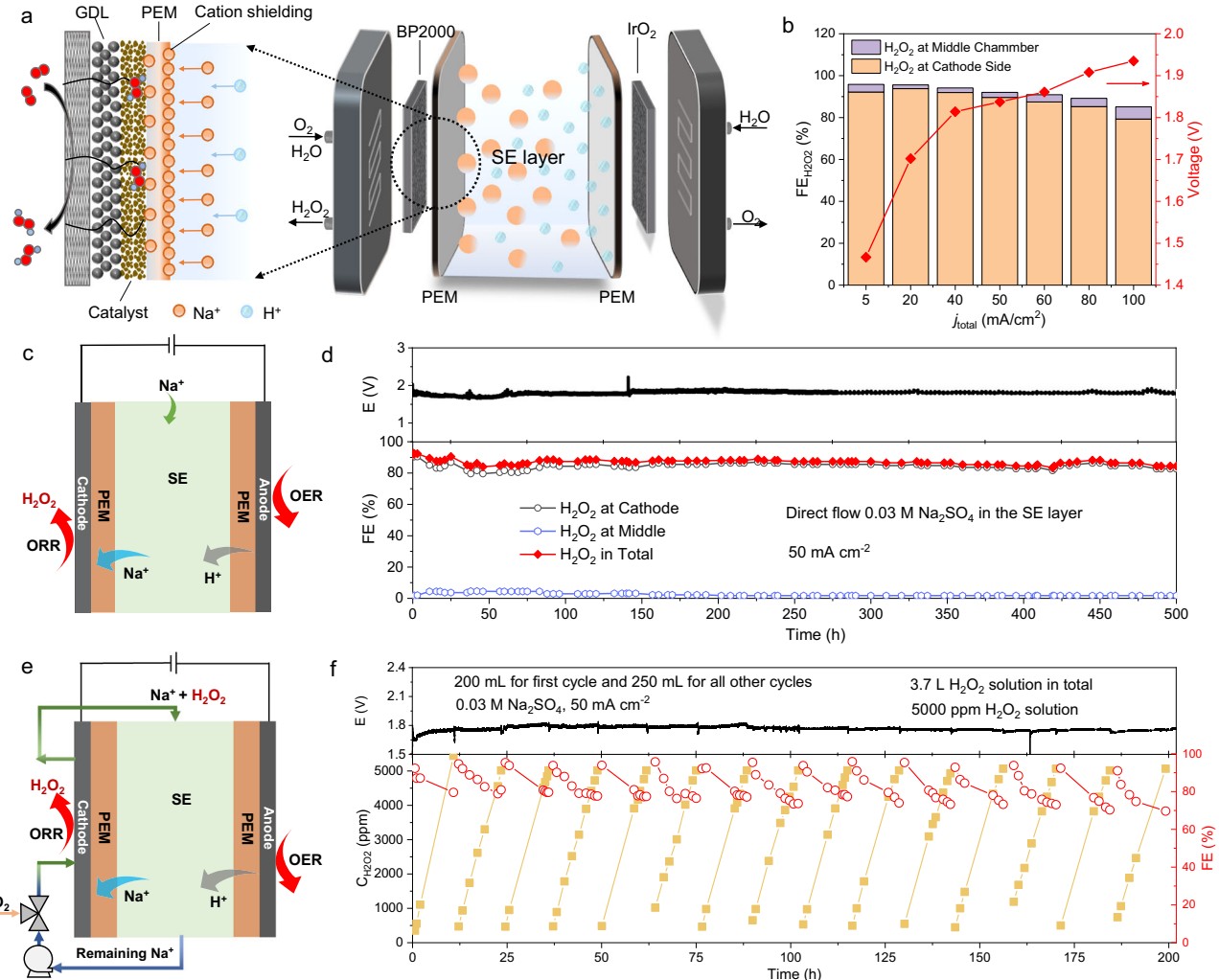

**Fig. 6 Continuous production of $H_2O_2$ solution using carbon catalyst in a SE cell with a double-PEM configuration. a** Schematic illustration of reducing $O_2$ to $H_2O_2$ in our SE cell with double-PEM configuration. The $O_2 + H_2O$/PEM//SE//PEM/$H_2O$ cell in which $O_2$ is reduced at the cathode side to form $H_2O_2$ and flowed out by $H_2O$ flow. The cations in the middle chamber cross over the PEM under an applied reduction potential and move to the cathode, protecting the catalyst surface for the production of $H_2O_2$. **b** The I–V curve and corresponding FEs for producing $H_2O_2$ using the SE cell with double-PEM configuration through flowing 0.03 M $Na_2SO_4$ in the middle chamber. The concentration of $Na_2SO_4$ can be varied. **c, d** The schematic illustration and chronopotentiometry stability test of the SE cell with double-PEM configuration by directly flowing 0.03 M $Na_2SO_4$ solution in the middle chamber at 50 mA·cm$^{-2}$ current density. The flow rate of $Na_2SO_4$ solution is 2.7 mL min$^{-1}$. The oxygen gas (flow rate 180 sccm) and DI water (flow rate 10.8 mL min$^{-1}$) are mixed and flowed into the cathode to producing $H_2O_2$ solution. DI water with flow rate of 2.7 mL min$^{-1}$ was circulated at the anode side. **e, f** The schematic illustration and chronopotentiometry stability test of the SE cell for practically producing 5000 ppm $H_2O_2$ solution. The volume of $Na_2SO_4$ stock solution in the first cycle is 200 mL, and the other 14 cycles hold 250 mL. The SE cell can produce around 3.7 L 5000 ppm $H_2O_2$ solution in 15 cycles for more than 200 h. The liquid flow rate is 4.5 mL min$^{-1}$ and the $O_2$ gas flow rate is 140 sccm. The total geometric area of the flow field in the cathode of our SE cell is 4 cm$^2$.

$H_2O_2$ solution was achieved in 17 h by circulation of the 50 mL water at the cathode side (Fig. S30). Based on the above promising $H_2O_2$ activity, selectivity and especially the durability, and since all the reactor components, including catalysts, membrane, and the polymer SE are all commercially available, our PEM-based $H_2O_2$ SE cell with cation promotion effect has a great potential for future's practical applications.

To conclude, we presented a cation-regulated interfacial engineering approach to improve the catalytic performance of $O_2$ reduction to $H_2O_2$ at industrial-relevant rates in strong acidic media. By adding a small number of alkali metal cations in acid solutions, the selectivity and stability of $H_2O_2$ generation using commercial carbon black catalyst can be dramatically improved, especially under large ORR current densities (over 400 mA cm$^{-2}$). Modeling of reaction and local environment suggest that the cations could preferentially be attracted to the catalyst/electrolyte

interface, showing a "shielding effect" to squeeze out the catalyst/ electrolyte interfacial protons and thus prevent further reduction of generated $H_2O_2$ to water. A double-PEM-based reactor was further developed for continuous production of $H_2O_2$ solution. By using only 0.03 M $Na_2SO_4$ as the cation source, a promoted $H_2O_2$ FE (~90%) and stability (>500 h) were achieved. In light of this performance, this would be a promising demonstration of the use of renewable electricity for the continuous generation of $H_2O_2$ through $O_2$ reduction at a more practical scale. This cation "shielding effect" could also be used in other electrocatalytic reactions such as selective $CO_2$ reduction into fuels and chemicals or $N_2$ reduction into ammonia.

## Experiment

*Materials.* All chemicals including lithium sulfate ($Li_2SO_4$), sodium sulfate ($Na_2SO_4$), potassium sulfate ($K_2SO_4$), caesium

sulfate ($Cs_2SO_4$), perchloric acid, sulfuric acid, and Nafion per-fluorinated resin solution (527084-25 mL) were purchased from Sigma Aldrich. $H_2O_2$ solution (35 wt%) was purchased from Merck & Co. Vulcan XC-72 were purchased from Fuel Cell Store. The conductive carbon black BP2000 was purchased from Cabot Corporation. Millipore water (18.2 MΩ·cm) was used throughout all experiments.

*Preparation of electrodes.* Typically, 40 mg conductive carbon black (BP2000) and 80 μL of Nafion (527084-25 mL) binder solution was mixed with 4 mL of 2-proponal (Sigma-Aldrich) and 1 ml methanol. After sonication in ice water for 30 min, the obtained homogeneous ink was air-brushed onto a $5 \times 5$-cm$^2$ gas diffusion layer (GDL, Sigracet 28 BC, Fuel Cell Store) electrode at room temperature. Then the prepared electrode was dried in a vacuum at room temperature for 24 h before use. The procedure for preparation electrodes with other catalysts is same as that of carbon black BP2000. The reduced graphene oxide (rGO) catalyst was pretreated using HCl and acetone to remove impurities before making the catalyst ink.

*Activation of the Nafion-117 membrane.* The proton exchange membrane (PEM, Nafion-117) was purchased from Fuel Cell Store. The Nafion-117 membrane was pre-treated with 5% (v/v) $H_2O_2$ for 1 h at 80 °C and 10% (v/v) $H_2SO_4$ for 1 h at 80 °C before assembling a cell.

*Materials characterization.* The scanning electron microscopy (SEM) was performed on an FEI Quanta 400 field emission scanning electron microscope. BET surface area analysis was performed using Quantachrome Autosorb-iQMP/Kr BET Surface Analyzer.

*Electrochemical test in flow cell.* The electrochemical $H_2O_2$ generation was conducted at 25 °C by using a conventional flow cell with a typical three-electrode setup, and the electrochemical response was recorded by using a BioLogic VMP3 workstation. The cathode and $IrO_2$ anode (Fuel Cell Store) were placed on opposite sides of two 0.5 cm thick PTFE plates with 0.5 cm * 2 cm channels. The catalyst layers faced the flowing liquid electrolyte, and the geometric surface area of the catalyst was 1 cm$^2$. A Nafion-117 film was sandwiched by the two PTFE plates to separate the chambers. At the cathode side, 30 sccm humidified $O_2$ was supplied through a titanium gas flow chamber, and a catholyte containing 0.1 M $H_2SO_4$ and cations flowed into the cathode chamber. The catholyte flow rate of 1.8 mL min$^{-1}$ was controlled by a syringe pump. The pH value of the catholyte was determined by an Orion 320 PerpHecT LogR Meter (Thermo Scientific). At the anode side, 0.1 M $H_2SO_4$ anolyte was circulated with a flow rate of 1.8 mL min$^{-1}$ for $O_2$ evolution reaction as the counter electrolyte. A saturated calomel electrode (SCE, CH Instruments) was employed as the reference electrode. All potentials measured against SCE were converted to the reversible hydrogen electrode (RHE) scale using $E_{RHE} = E_{SCE} + 0.241 V + 0.0591 \times pH$. The resistance (Rs) of the catalytic system was determined by poten-tiostatic electrochemical impedance spectroscopy (PEIS) at fre-quencies ranging from 0.1 Hz to 200 kHz. All the measured potentials using the three-electrode flow cell setup were manually 85% compensated unless stated otherwise.

*Solid state electrolyte cell with double-PEM configuration.* The continuous electrosynthesis of $H_2O_2$ was conducted using a solid electrolyte (SE) cell with a sandwiched double-PEM configura-tion. The cell configurations and the production setup are illu-strated in Fig. 6a and Fig. S25. The cathode side was supplied with an oxygen/water mixture of 180 sccm of $O_2$ gas and

10.8 mL min$^{-1}$ of DI water. The gas flow rate was controlled by a mass flow meter (MFC) and the water flow rate was controlled by a syringe pump. The flow rate of $H_2O_2$ product at the outlet was calibrated using a measuring cylinder. The fast water flow in the gas/liquid mixture through the cathode chamber is beneficial for bringing out the generated $H_2O_2$ molecules and decreasing the further electroreduction of $H_2O_2$. In the middle chamber, the styrene-divinylbenzene sulfonated copolymer Dowex 50WX8 hydrogen form (Sigma–Aldrich) cation conductor was employed as the SE. A solution containing $H_2SO_4$ and/or $Na_2SO_4$ flowed into the SE layer controlled by a syringe pump. The anode side was circulated with 0.1 M $H_2SO_4$ at 2.7 mL min$^{-1}$. All the mea-sured potentials using a two-electrode setup were manually 100% compensated unless stated otherwise.

*Bath synthesis of 5000 ppm $H_2O_2$ Solution using double-PEM cell.* The bath electrosynthesis of 5000 ppm $H_2O_2$ was conducted using double-PEM cell configuration (as shown in Figs. 6a and S25). A certain volume of 0.03 M $Na_2SO_4$ solution (200 mL for the first cycle, and 250 mL for other cycles) was supplied into the middle SE layer with flow rate of 4.5 ml min$^{-1}$. The outlet of the middle chamber was mixed with 140 sccm $O_2$ gas and then supplied to the cathode side for producing $H_2O_2$. The cathode outlet containing $H_2O_2$ and remaining $Na_2SO_4$ was then circu-lated back to the middle SE chamber for continually running of the cell. Once the accumulated $H_2O_2$ concentration reached around 5000 ppm, the cell was flushed with fresh 0.03 M $Na_2SO_4$ for 10 min to remove residue $H_2O_2$, and another bottle of fresh 0.03 M $Na_2SO_4$ (250 mL) was used to start a new batch.

*Determination of the $H_2O_2$ concentration.* The concentration of the generated $H_2O_2$ was determined through a titration process. After electrolysis, the as-produced $H_2O_2$ solution was collected and evaluated using the standard potassium permanganate (0.1 N KMnO$_4$ solution, Sigma–Aldrich) titration process, according to the following equation:

$$2MnO_4^- + 5H_2O_2 + 6H^+ \rightarrow 2Mn^{2+} + 5O_2 + 8H_2O \quad (1)$$

The sulfuric acid (1 M $H_2SO_4$) was used as the H$^+$ source. The FE for $H_2O_2$ production is calculated using the following equation:

$$FE = \frac{generated\ H_2O_2\left(mol\ L^{-1}\right) \times 2 \times 96485\left(C\ mol^{-1}\right) \times flow\ rate\left(mL\ s^{-1}\right)}{j_{total}(mA)}$$
$$\times 100\ (maximum\ 100\%)$$
$$(2)$$

*Electrochemical $H_2O_2$ dissociation.* The electrochemical $H_2O_2$ dissociation was conducted in a customized gas-tight H-type glass cell at 25 °C. Before the experiment, the glass cell was carefully cleaned by boiling the cell in a mixture of $H_2SO_4$: $H_2O_2$ (3:1) for 1 h. After being thoroughly cleaned by DI at room temperature, the cell was further boiled in DI water for another 1 h to totally remove $H_2O_2$ residual.

The electrochemical $H_2O_2$ dissociation was conducted with a BioLogic VMP3 workstation. The cathode electrode was prepared by spray coating carbon black (BP2000) on a GDL (Sigracet 28 BC, Fuel Cell Store), and the anode electrode was a carbon rod. The cathode electrode was fixed using a gold-coated clip and the exposed geometric surface area of each electrode was 1 cm$^2$. Since the traditional clip made of iron can be easily dissolved to Fe$^{2+}$/Fe$^{3+}$ by acid and may contribute to the $H_2O_2$ dissociation, the gold-coated clip is necessary to avoid the dissolution of the iron clip during the process. The working and counter electrodes were parallel and separated by a clean PEM. The mixture of 0.2 M $H_2O_2$ + 0.1 M $H_2SO_4$ was used as catholyte, 0.1 M $H_2SO_4$ was used as anolyte, and

each volume of the electrolyte was 25 mL. A gas dispersion frit was used at the cathode chamber to provide vigorous electrolyte mixing. The cathode chamber was supplied with Ar gas (99.999% Praxair) at a rate of 20 sccm for 30 min before the electrochemical measurements. During electrolysis, continuous Ar flow was supplied throughout the experiment and the gas outlet was connected to a gas chromatograph (GC, Shimadzu GC-2014 GC) for detection of the $H_2$ gas. The $H_2$ amount was quantified by a thermal conductivity detector. After the electrochemical decomposition, the amount of $H_2O_2$ remaining was determined by using the standard potassium permanganate titration process.

*Theoretical simulation.* The Vienna Ab initio Simulation Package (VASP)[40,41], together with the VASPsol patch[42], was employed to perform slow-growth calculation. The constant potential along the MD track is realized by adjusting the number of electrons on-the-fly, as described in our previous work[35]. Perdew-Burke-Ernzerhof (PBE) functional[43] together with D3 van der Waals correction[44] were employed in most of the calculations. The cutoff energy of the plane-wave basis is 400 eV in the relaxation while 300 eV was used in the MD simulations. Gamma-only MD calculations were done for the thick model in Fig. 5a, b (water layers equivalent to 6 ice layers), while $3 \times 3 \times 1$ Gamma-centered k-mesh was used in MD simulations using the thinner defect-graphene model shown in Fig. S17. One proton is added into the 45 $H_2O$ molecules to simulate the pH = 0 condition. We choose the bond length (*-O or O-OH) to be the reaction coordinate ($\varepsilon$). Time step in MD was set to be 0.5 femtoseconds, and the $\partial \varepsilon$ in slow-growth method[35] was set to be 0.0004 Å. A Nose-Hoover thermostat[45] was used to keep temperature (statistically) constant at 300 K. The proton is determined as the H that is farthest from the central O among two other H atoms of a $H_3O^+$ for snapshots that are evenly distributed along the AIMD track.

## Data availability

The data that support the findings of this study are available from the corresponding authors upon reasonable request.

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

## Acknowledgements

This work was supported by the Robert A. Welch Foundation (grant no. C-2051-20200401) and the David and Lucile Packard Foundation (grant no. 2020-71371). Y.L. acknowledges the support by NSF (Grant No. 1900039), ACS PRF (60934-DNI6), and the Welch Foundation (Grant No. F-1959-20210327). The calculations used computational resources at XSEDE, TACC, Argonne National Lab, and Brookhaven National Lab. X.Z. acknowledges the support by the Fondazione Oronzio e Niccolò De Nora in Applied Electrochemistry.

## Author contributions

X.Z. and X.H.Z. contributed equally. X.Z. and H.W. conceived the project and designed the experiments. X.Z., P.Z., Z.A., and Z.W. perform the experimental study. X.H.Z. and Y.L. performed the theoretical study. X.Z., X.H.Z., Y.L., and H.W. wrote the manuscript with support from all authors.

## Competing interests

A patent application has been filed based on this study.
