## [Peer review file · Nature Communications]

REVIEWER COMMENTS

Reviewer #1 (Remarks to the Author):

Zhang & Zhao et al. describes a practical way of electrochemically producing H₂O₂. High current density operation with high H₂O₂ selectivity with low cost cathode catalyst was achieved by introducing double (or triple) PEM layered electrochemical cell with Na⁺ ion introduced as a promoter to boost selectivity towards H₂O₂.

Most of the recent works on electrochemical H₂O₂ production focuses on catalysts development in alkaline electrolyte. However, practical device development is limited by alkaline electrolyte membrane not by catalysts. And there is little room for improvement for catalysts in alkaline environment. In contrast, ORR to H₂O₂ in acidic environment is more potent, at least in my opinion, where we already have viable polymer electrolyte membrane option. Development of catalysts and designing a suitable device are challenges exist in PEM based H₂O₂ production.

This work successfully tackles device design and operation strategies for PEM based H₂O₂ production by introducing double (or triple) PEM layered electrochemical cell with continuous Na⁺ supply. This work is timely and requires attention from the field from an engineering perspective.

However, this work lacks scientific explanation of the phenomenon observed. I think this work will be appropriate for the publication in Nat. Comm with with major revision addressing these problems.

Major comments

1. Na⁺ not only increases selectivity but also activity and stability. However, authors only focus on the selectivity. And discussions should also include activity and stability enhancement.
2. AIMD simulation is difficult to understand.
 - a) What is the difference between figure 4a and 4b? And why Na⁺ have continuous values while H⁺ values are discrete?
 - b) Authors state that Na⁺ drifted towards the surface(P11 L19), but it is not so obvious in Figure 4a, b. Two Na⁺ only slighted drifted (~1 angstrom) and 2nd Na⁺ did not drift towards the surface. In figure 4a, a slight drift of 1 Å is relevant?
 - c) BP2000 is defect rich and contains a lot of micro-pores.
(<https://www.sciencedirect.com/science/article/pii/S001346861830687X>) Using graphitic surface may be inappropriate.
 - d) There are two H⁺ and two Na⁺ in the AIMD model. But isn't proton concentration much higher than Na⁺ concentration in operating condition?
 - e)

3. What is the reason for selectivity degradation of BP2000 without Na⁺? Does carbon decompose at such negative potential?
4. In P13-L17,18, rates are compared for two very different reactions. I think reaction constants will be different for these two reactions.
5. It seems like that crossed-over Na⁺ concentration is almost same as H₂O₂ concentration (Fig S23). Do we need a mole of Na⁺ to produce a mole of H₂O₂? Then will it be relevant strategy in a real world device?
6. Why did authors use 50mA/cm² for long-term operation test for double-PEM configuration? 50mA/cm² seems low for practical application.
7. Double-PEM configuration without Na⁺ possess higher selectivity than single-PEM configuration with Na⁺ in the anolyte. (Figure S20 vs S24) It seems like device design is also very important.

Minor comments

1. P13-L4 : absent → present
2. P5-L10 : H₂O₂ selectivity and FE are used confusingly.
3. Figure 5 c,e is confusing where cathodes and anodes positions are flipped from 5a

Reviewer #2 (Remarks to the Author):

This work presented an interesting point about the influence of alkali metal cations on the electrocatalytic H₂O₂ production in acidic media. The “shielding effect” induced by alkali metal cations is supposed to squeeze away the catalyst/electrolyte interfacial protons and thus prevent further reduction of generated H₂O₂ to water, thus achieving the high selectivity and stability. The authors performed various characterization techniques combined with calculations, and presented sufficient evidence to support their claims. Overall, this work is novel and can be published after minor revisions. Below are some comments that the authors could consider for the further improvement of this work:

1. The pH value is adjusted to about 1 for the test. What would the catalytic performance be if the pH value further decreases or increases? For example, considering pH = 0 is used for the simulations, would the catalytic selectivity change significantly if using this pH value?
2. Would the H₂O₂ generation be always promoted if further more Na₂SO₄ is added?

3. The authors also explored other alkali metal ions in group IA and similar enhanced H₂O₂ selectivity was observed. What about the influence of other metal ions for the H₂O₂ generation, such as Mg²⁺ and Al³⁺?
4. Fig. 3b shows the enhanced performance in 0.1 M H₂SO₄ other than 0.1 M H₂SO₄ + 0.05M Na₂SO₄, which is contradictory with the description in the main text. The authors should carefully check this.
5. In the I-V curve, the authors used the *j*_{total}. It's recommended to provide a JH₂O₂-V curve for the better comparison.
6. The authors could refer to some relevant work on the catalyst design for H₂O₂ generation. Such as Adv. Mater. 2021, 2104891.

Reviewer #3 (Remarks to the Author):

The authors reported the addition of alkaline cations in electrolyte can significantly promote the production of H₂O₂. The promoted H₂O₂ FE over 80% and the stability over 5000 hours. The ab initio MD simulation examined the mechanism and found that cations can repel proton in electrolyte, shielding effect, to avoid the overoxidation reaction of H₂O₂ (forming H₂O); thus, improve the production of H₂O₂. Practical reactor for H₂O₂ generation is also demonstrated. The comments are listed below.

1. It is suggested that the authors can separate the scheme in Fig. 1a with other Fig. 1b-1d to smooth the description. Fig. 1a mentioned in Introduction, while other described in the next section.
2. The authors should specifically emphasize the difference between present ORR forming H₂O₂ and another important ORR forming H₂O, which is cathodic reaction in the promising PEMFC and has been extensively studied.

For example, the present ORR desires the formation of H₂O₂ and try to avoid the production of H₂O; however, the ORR in PEMFC application is completely opposite. Dose that mean the “bad” results in the present study can be a “good” one in the ORR in PEMFC? The authors need to clarify the difference between those two ORR.

3. The authors claimed that “higher concentration of Na⁺ can maintain larger 2e—ORR currents without sacrificing the H₂O₂ selectivity”. Their results also showed that the highest concentration (0.1 M Na₂SO₄) shows the best H₂O₂ selectivity. Does that mean the selectivity can be further improved with even higher concentration (ex. 0.2 M)? Is there the optimized concentration to achieve the best selectivity (e.g. volcano plot)?
4. The authors employed DFT to the mechanism study and found that the enhanced H₂O₂ selectivity corresponds to the repulsion between cation (Na⁺) and proton (H⁺) to ease the further protonation of

H₂O₂ forming the unwanted H₂O. On the other hand, the experimental results found that other IA cations (Li⁺, K⁺ and Cs⁺) also showed similar result.

There are some inconsistencies between those DFT and experimental results. According to DFT results, different sized cations should induce different repulsion to altered the H₂O₂ selectivity to some degree; however, the experiments observed the same promotion of all the cations (Fig. 2d-f). The authors can try with other cations with higher charges (e.g. Ca²⁺, Mg²⁺...) or simulate with all the experimental (1A) cations to clarify it. According to the experimental results, colligative property might be the more suitable mechanism in the present work.

5. The experimental results (Fig. 3) found that CNT, rGO and Zn-N-C are suitable catalysts for the present application. Also, their catalytic performance (production rate ~6 mmol/cm²/h) are better than carbon black (BP2000) (~3 mmol/cm²/h). It is suggested the authors to use the better CNT, rGO or Zn-N-C to run to the latter tests of continuous production of H₂O₂ (Fig. 5) to achieve a better performance.

The authors reported the addition of alkaline cations in electrolyte can significantly promote the production of H_2O_2 . The promoted H_2O_2 FE over 80% and the stability over 5000 hours. The ab initio MD simulation examined the mechanism and found that cations can repel proton in electrolyte, shielding effect, to avoid the overoxidation reaction of H_2O_2 (forming H_2O); thus, improve the production of H_2O_2 . Practical reactor for H_2O_2 generation is also demonstrated. The comments are listed below.

1. It is suggested that the authors can separate the scheme in Fig. 1a with other Fig. 1b-1d to smooth the description. Fig. 1a mentioned in Introduction, while other described in the next section.
2. The authors should specifically emphasize the difference between present ORR forming H_2O_2 and another important ORR forming H_2O , which is cathodic reaction in the promising PEMFC and has been extensively studied.
For example, the present ORR desires the formation of H_2O_2 and try to avoid the production of H_2O ; however, the ORR in PEMFC application is completely opposite. Does that mean the “bad” results in the present study can be a “good” one in the ORR in PEMFC? The authors need to clarify the difference between those two ORR.
3. The authors claimed that “higher concentration of Na^+ can maintain larger $2e^-$ ORR currents without sacrificing the H_2O_2 selectivity”. Their results also showed that the highest concentration (0.1 M Na_2SO_4) shows the best H_2O_2 selectivity. Does that mean the selectivity can be further improved with even higher concentration (ex. 0.2 M)? Is there the optimized concentration to achieve the best selectivity (e.g. volcano plot)?
4. The authors employed DFT to the mechanism study and found that the enhanced H_2O_2 selectivity corresponds to the repulsion between cation (Na^+) and proton (H^+) to ease the further protonation of H_2O_2 forming the unwanted H_2O . On the other hand, the experimental results found that other IA cations (Li^+ , K^+ and Cs^+) also showed similar result.
There are some inconsistencies between those DFT and experimental results. According to DFT results, different sized cations should induce different repulsion to altered the H_2O_2 selectivity to some degree; however, the experiments observed the same promotion of all the cations (Fig. 2d-f). The authors can try with other cations with higher charges (e.g. Ca^{2+} , Mg^{2+} ...) or simulate with all the experimental (1A) cations to clarify it. According to the experimental results, colligative property might be the more suitable mechanism in the present work.
5. The experimental results (Fig. 3) found that CNT, rGO and Zn-N-C are suitable catalysts for the present application. Also, their catalytic performance

(production rate ~ 6 mmol/cm²/h) are better than carbon black (BP2000) (~ 3 mmol/cm²/h). It is suggested the authors to use the better CNT, rGO or Zn-N-C to run to the latter tests of continuous production of H₂O₂ (Fig. 5) to achieve a better performance.

Haotian Wang
Assistant Professor
Phone: 713-348-7221
Email: htwang@rice.edu

ChBE Department
Rice University
6100 Main Street
Houston, TX 77005

March 23, 2022

Manuscript Number: NCOMMS-21-50208

Title: "Electrochemical oxygen reduction to hydrogen peroxide at practical rates in strong acidic media"

Authors: Xiao Zhang, Xunhua Zhao, Peng Zhu, Zachary Adler, Zhen-Yu Wu, Yuanyue Liu, Haotian Wang

Corresponding authors: Xiao Zhang, Yuanyue Liu and Haotian Wang

Response to reviewers' comments:

We thank the reviewers and editor for the constructive comments which have helped us to greatly improve our research and the quality of our manuscript. We have now included additional analysis and characterizations, and performed substantial experiments and simulations to fully address the reviewers' concerns and suggestions. Below, we address the points raised by reviewers one by one.

Reviewer 1

Zhang & Zhao et al. describes a practical way of electrochemically producing H_2O_2 . High current density operation with high H_2O_2 selectivity with low-cost cathode catalyst was achieved by introducing double (or triple) PEM layered electrochemical cell with Na^+ ion introduced as a promoter to boost selectivity towards H_2O_2 . Most of the recent works on electrochemical H_2O_2 production focuses on catalysts development in alkaline electrolyte. However, practical device development is limited by alkaline electrolyte membrane not by catalysts. And there is little room for improvement for catalysts in alkaline environment. In contrast, ORR to H_2O_2 in acidic environment is more potent, at least in my opinion, where we already have viable polymer electrolyte membrane option. Development of catalysts and designing a suitable device are challenges exist in PEM based H_2O_2 production. This work successfully tackles device design and operation strategies for PEM based H_2O_2 production by introducing double (or triple) PEM layered

electrochemical cell with continuous Na^+ supply. This work is timely and requires attention from the field from an engineering perspective. However, this work lacks scientific explanation of the phenomenon observed. I think this work will be appropriate for the publication in Nat. Comm with major revision addressing these problems.

Response: We appreciate the reviewer's strong support of our work for publication, as well as the suggestions which have substantially improved the quality of our manuscript. In the present version of the manuscript, we have addressed all questions raised by all three reviewers regarding both the experimental and theoretical analysis, which has greatly improved the depth and rigor of this work.

Major comments

Comment 1: Na^+ not only increases selectivity but also activity and stability. However, authors only focus on the selectivity. And discussions should also include activity and stability enhancement.

Response: We appreciate the reviewer's important suggestion. In our simulation study, we found that the solvated alkali metal cations, compared to concentrated protons in acids, could preferentially be attracted to the catalyst/electrolyte interface and squeeze out local protons during ORR reaction. Therefore, the local concentration of Na^+ is much higher than that of H^+ , which gives rise to a local alkaline environment during the ORR process. At alkaline conditions, the carbon-based catalyst typically shows higher activity and better stability compared to acidic conditions (*Nature Catalysis* **2018**, *1*, 282; *Nature Catalysis* **2018**, *1*, 156; *Nature Materials* **2020**, *19*, 436). Therefore, we suppose the improved activity and stability are induced by the alkalization of the local environment induced by the cation additives. The related description has been included and highlighted in red in the revised version of the manuscript on Page 10, which is also shown below for your reference:

"We suppose the improved activity and stability are induced by the alkalization of local environment induced by the cation additives during ORR".

Comment 2: AIMD simulation is difficult to understand.

a) What is the difference between figure 4a and 4b? And why Na^+ have continuous values while H^+ values are discrete?

Response: We thank the reviewer for raising this important point. As already shown in the

Figs. 5a and 5b (copied below), the Fig. 5a combines two separated MD runs, and shows when only Na^+ cations *or* H^+ are presented in separated MD runs, while the Fig. 5b shows when both Na^+ *and* H^+ are presented. To make it clearer, we further clarified the difference in the caption of Figs. 5a and 5b accordingly (shown below).

Regarding the second question (why Na^+ have continuous values while H^+ values are discrete), the position of Na^+ is straightforward to know, while whether a H being a proton or part of a neutral H_2O molecule is not always easy to determine. We define the proton as the H that is furthest from the central O among two other H atoms of a H_3O^+ . This is done manually for evenly distributed snapshots along the MD track. Consequently, the points for H^+ are discrete. We thank the reviewer's question, and we clarified this in the Supporting Information as "**The proton is determined as the H that is farthest from the central O among two other H atoms of a H_3O^+ for snapshots that are evenly distributed along the AIMD track.**"

Fig. 5 (a, b) Evolution of the position (surface normal direction "z") of Na^+ ions (and/or) H^+ after positioning them in the water layers.

b) Authors state that Na^+ drifted towards the surface (P11 L19), but it is not so obvious in Fig. 4a, b. Two Na^+ only slightly drifted (~ 1 angstrom) and 2nd Na^+ did not drift towards the surface. In Fig. 4a, a slight drift of 1 Å is relevant?

Response: We thank the reviewer's question. The drifts of Na^+ are slightly less than 2 Å in Fig. 5a. In Fig. 5b the drift of the 1st Na^+ is more than 3 Å towards the surface until it reaches a similar z value of the 2nd Na^+ . The 2nd Na^+ is already close (1.5 Å to the surface) to the surface, so there is no space for its further drift. Since the net (negative) charge of the surface is more than 3, Fig. 5b is closer to the concentration of the total cations at the reaction potential, while Fig. 5a is for comparison purposes. Furthermore, the relatively small drift is also limited by the relatively

small model that we can afford for long AIMD runs. Nevertheless, there is no doubt about the qualitative difference between the drifts of Na^+ and H^+ , so we believe the drift values are large enough.

c) BP2000 is defect rich and contains a lot of micro-pores. (<https://www.sciencedirect.com/science/article/pii/S001346861830687X>) Using graphitic surface may be inappropriate.

Response: We thank the reviewer's question. As shown in Fig. 5e (copied below), we actually used a defected and pored surface as our catalyst model. To make it clearer, we further clarify this in the Supporting Information (section of "Theoretical Simulation") as "using the thinner defect-graphene model shown in Fig. S17".

Fig. 5. (e1-e3) Initial state, transition state and final state of the H_2O_2 decomposition in neutral solution. C: grey; O: red; H: white; Proton: cyan.

d) There are two H^+ and two Na^+ in the AIMD model. But isn't proton concentration much higher than Na^+ concentration in operating condition?

Response: We appreciate the reviewer's question. As we have shown, Na^+ is likely to be attracted towards the surface and repel away H^+ . Even though the Na^+ concentration in bulk solution is much lower than H^+ , the surface concentration of H^+ is not necessarily higher than Na^+ .

Comment 3: What is the reason for selectivity degradation of BP2000 without Na^+ ? Does carbon decompose at such negative potential?

Response: We thank the invaluable comments. In pure 0.1 M H_2SO_4 , the H_2O_2 selectivity of BP2000 gradually decreases with operation duration. We hypothesize that it is due to the gradual accumulation of local H^+ over the continuous reaction, and H_2O_2 is reduced to H_2O in the strong

acidic condition with high H^+ concentration, but not due to the decomposition of carbon catalyst. In acidic solutions, the local H^+ gradually accumulate to high concentrations along with the ORR reaction time. Under a high concentration of local H^+ , when O_2 is reduced to H_2O_2 in its first place, those generated H_2O_2 at the electrode surface could be easily further reduced to H_2O coupling two electrons and two local protons, which decrease the H_2O_2 selectivity. The carbon catalyst was not decomposed at such negative potential, and is not the reason for selectivity degradation. One piece of direct evidence is that the Na^+ additives can recover the H_2O_2 selectivity to a high value even if it is significantly decreased under pure acids. As shown in Fig. S29 in the revised version (copied below), in pure 0.1 M H_2SO_4 , the H_2O_2 selectivity at 200 mA/cm² rapidly dropped to less than 10% within 6 hours, while as a sharp contrast, the H_2O_2 FE was quickly recovered to high value with the addition of cations. The observation indicates that the carbon catalyst still works well, and it is not the reason for selectivity degradation in pure acids.

Fig. S29. The effect of alkali metal cations toward the H_2O_2 production stability in the double-PEM SE cell. Without cations, the FE of H_2O_2 continuously declines to less than 10% within 6 h, while the addition of 0.03 M Na_2SO_4 in the middle chamber improves the FE of H_2O_2 . The FE of H_2O_2 production with 0.03 M Na_2SO_4 in the middle chamber is well maintained, suggesting a continuous and stable generation of H_2O_2 solutions.

Comment 4: In P13-L17,18, rates are compared for two very different reactions. I think reaction constants will be different for these two reactions.

Response: We appreciate the reviewer's excellent point. If we understand correctly the reviewer

means the frequency factor A in Arrhenius equation ($k = Ae^{-E_A/RT}$) is different for the two reactions. However, as we know this frequency factor is linearly correlated to the reaction rate, while the activation energy E_A , which is relevant to the Gibbs free energy we calculated, is exponentially correlated to the reaction rate, we believe the free energy difference usually dominates the difference in reaction rates. Using this free energy barrier to have a rough estimation of the reaction rates has been widely adopted in DFT simulations in electrocatalysis. In addition, considering that the $2e^-$ pathway is finished after the breaking of $*-O$ while $4e^-$ pathway takes more steps ($*O \rightarrow *OH$ and removal of $*OH$) to finish, the difference between of $2e^-$ path vs. $4e^-$ path rates is likely to be more significant than our estimation ($2e^-/4e^- \sim 3.36$ times). Therefore, our analysis is likely to be reasonable.

Comment 5: It seems like that crossed-over Na^+ concentration is almost same as H_2O_2 concentration (Fig S23). Do we need a mole of Na^+ to produce a mole of H_2O_2 ? Then will it be relevant strategy in a real world device?

Response: We appreciate the reviewer's important question. In our solid electrolyte reactor, the Na^+ in the middle chamber penetrate the PEM closest to the cathode side to modify the catalytic interface for H_2O_2 production. If we assume that the Na^+ crossover ratio is 100 % and H^+ crossover is 0%, in principle, 2 mol Na^+ is needed to produce 1 mol H_2O_2 based on the charge balance. However, as shown in Fig. S27a (copied below), the Na^+ crossover ratio gradually decreased with increased current density. This is reasonable because high current densities need more cation flux, and the low concentrations of Na^+ in the middle layer cannot make up the ion flux, therefore protons start to crossover to make up the charge. This as a result will help to increase the H_2O_2 to crossover Na^+ ratio (Fig. S27b), but the cation shielding effect starts to decrease due to stronger proton flux. As shown in Fig. 6b, under higher current densities, the H_2O_2 selectivity starts to decrease due to the lower cross ratio of Na^+ ions compared to protons.

In practical use, we prefer to lower the concentrations of middle layer Na^+ ions, but the H_2O_2 selectivity could be harmed. To address this dilemma, we operated the cell by circulating the cation solution from SE layer into the cathode side, and then back to SE layer for H_2O_2 accumulation (Figs. 6e and 6f, copied below). By doing so, we can allow sufficient Na^+ crossover to the cathode to ensure a good shielding effect, at the same time the crossovered Na^+ can be recycled back without the need of continuous addition of extra Na^+ . As shown in Fig. 6f, the H_2O_2 concentration

continuously increased to ~ 5000 ppm in about 13 hours. For each operation cycle, the H_2O_2 concentration can reach to ~ 0.15 M, which is 5 times of the Na_2SO_4 additive in the final product. Therefore, the reactor does not need 2 moles of Na^+ to produce one mole of H_2O_2 .

Fig. S27. (a) The Na^+ crossover ratio through the cathodic PEM in the SE cell by using 0.03 M Na_2SO_4 in the middle chamber. (b) The correlation of the crossover of Na^+ with the production rate of H_2O_2 . The concentration and concentration ratio in Fig. S27b are based on the mass concentration of H_2O_2 and Na^+ tested from ICP.

Fig. 6. a) Schematic illustration of reducing O_2 to H_2O_2 in our SE cell with double-PEM configuration. The $\text{O}_2+\text{H}_2\text{O}/\text{PEM}/\text{SE}/\text{PEM}/\text{H}_2\text{O}$ cell in which O_2 is reduced at the cathode side to form H_2O_2 and flowed out by H_2O flow. The cations in the middle chamber cross over the PEM under an applied reduction potential and move to the cathode, protecting the catalyst surface for

the production of H₂O₂. **b)** The I-V curve and corresponding FEs for producing H₂O₂ using the SE cell with double-PEM configuration through flowing 0.03 M Na₂SO₄ in the middle chamber. The concentration of Na₂SO₄ can be varied. **e, f)** The schematic illustration and chronopotentiometry stability test of the SE cell for practically producing 5000 ppm H₂O₂ solution. The volume of Na₂SO₄ stock solution in the first cycle is 200 mL, and the other 14 cycles hold 250 mL. The SE cell can produce around 3.7 L 5000 ppm H₂O₂ solution in 15 cycles for more than 200 h. The liquid flow rate is 4.5 mL min⁻¹ and the O₂ gas flow rate is 140 sccm. The total geometric area of the flow field in the cathode of our SE cell is 4 cm².

Comment 6: Why did authors use 50mA/cm² for long-term operation test for double-PEM configuration? 50mA/cm² seems low for practical application.

Response: We appreciate the reviewer's important suggestion. There is a trade-off relationship between the H₂O₂ selectivity, current density, and cation concentration. To get a high H₂O₂ selectivity at high current densities during catalysis, more cations are needed. For example, when using only 0.03 M Na₂SO₄ as additive in a solid electrolyte reactor, the H₂O₂ FE will drop a little bit to 80% once the current density increases to 100 mA/cm². Nevertheless, we can also boost the FE by using higher concentration of Na₂SO₄ additive (for example, 0.05 M Na₂SO₄). Our target is to realize the practical usage of our reactor and our H₂O₂ product. For practical applications, small number of cations (salt) is more attractive, and especially valuable for on-site household production. Therefore, in our design, we tried to minimize the concentration of cations while maintaining a relatively high FE to make it more practical. We highlight this point in the revised manuscript on Page 16, which is also listed here: "Our target is to realize the high production rate of H₂O₂ while minimizing cations' consumption for practical demonstrations".

Comment 7: Double-PEM configuration without Na⁺ possess higher selectivity than single-PEM configuration with Na⁺ in the anolyte. (Figure S20 vs S24) It seems like device design is also very important.

Response: We thank the reviewer for raising this important comment. As the reviewer pointed that, the double-PEM configuration without Na⁺ in the middle chamber shows slightly higher selectivity than single-PEM configuration with Na⁺ in the anolyte (Fig. S24 vs. S28 in the revised version, also copied below). We suspect this is due to the different H⁺ flux distribution of the two reactors during electrocatalysis. In single-PEM reactor, the catalysts at anode and cathode are closely in touch with PEM membrane. During catalysis, a number of local H⁺ are generated at the

anode/membrane interface, which suppress the possibility of Na^+ transportation from the anolyte towards the cathode to regulate the interfacial environment of improved $2e^-$ -ORR. Moreover, we suspect that the H^+ generated at the membrane/anode interface is not uniformly distributed due to the ununiformed OER catalyst on the anode. There could be local hotspot for H^+ generation and thus results in ununiform H^+ flux towards the cathode/membrane interface, which will decrease the H_2O_2 FE. Unlike the single-PEM reactor, the cathode and anode catalyst in our solid electrolyte reactor are separated by a buffer SE layer. The generated H^+ from anode will first move into the middle solid electrolyte chamber and then move across the cathode PEM, during which the H^+ flux distribution will become uniform. This H^+ flux is driven by the electrical field to uniformly penetrate the PEM toward the cathode surface without forming hotspots, and thus giving rise to a slightly better H_2O_2 selectivity.

Fig. S24b. FE of H_2O_2 production using MEA with configurations of $\text{O}_2+\text{H}_2\text{O}/\text{BP}/\text{PEM}/\text{IrO}_2/0.05\text{M Na}_2\text{SO}_4$. **Fig. S28b.** The corresponding FE of production of H_2O_2 as a function of current densities.

Minor comments

1. P13-L4: absent → present

Response: We appreciate the reviewer's careful check. Revised as suggested, thank you for the careful reading of the manuscript. We also checked the whole manuscript and modified some typos.

2. P5-L10: H_2O_2 selectivity and FE are used confusingly.

Response: We appreciate the reviewer's important suggestion. To make it consistent, in the

manuscript, we revised most specific descriptions of "H₂O₂ selectivity" to "H₂O₂ FE" to indicate the electrochemical selectivity of H₂O₂.

3. Figure 5 c, e is confusing where cathodes and anodes positions are flipped from 5a.

Response: We appreciate the reviewer's important suggestion. We have changed the two figures as suggested. Moreover, to make it clearer, we also changed the color of PEM in Fig. 6c and 6e (revised version) using the similar color in Fig. 6a. We also use the same description, i.e., PEM, to indicate the membrane part. The revised figures are also shown below for your reference:

Fig. 6. Continuous production of H₂O₂ solution using carbon catalyst in a SE cell with a double-PEM configuration. **a**, Schematic illustration of reducing O₂ to H₂O₂ in our SE cell with double-PEM configuration. The O₂+H₂O/PEM//SE//PEM/H₂O cell in which O₂ is reduced at the cathode side to form H₂O₂ and flowed out by H₂O flow. The cations in the middle chamber cross over the PEM under an applied reduction potential and move to the cathode, protecting the catalyst surface for the production of H₂O₂. **b**, The I-V curve and corresponding FEs for producing H₂O₂ using the SE cell with double-PEM configuration through flowing 0.03 M Na₂SO₄ in the middle

chamber. The concentration of Na_2SO_4 can be varied. **c, d, The schematic illustration** and chronopotentiometry stability test of the SE cell with double-PEM configuration by directly flowing 0.03 M Na_2SO_4 solution in the middle chamber at $50 \text{ mA}\cdot\text{cm}^{-2}$ current density. The flow rate of Na_2SO_4 solution is 2.7 mL min^{-1} . The oxygen gas (flow rate 180 sccm) and DI water (flow rate 10.8 mL min^{-1}) are mixed and flowed into the cathode to producing H_2O_2 solution. DI water with flow rate of 2.7 mL min^{-1} was circulated at the anode side. **e, f, The schematic illustration** and chronopotentiometry stability test of the SE cell for practically producing 5000 ppm H_2O_2 solution. The volume of Na_2SO_4 stock solution in the first cycle is 200 mL, and the other 14 cycles hold 250 mL. The SE cell can produce around 3.7 L 5000 ppm H_2O_2 solution in 15 cycles for more than 200 h. The liquid flow rate is 4.5 mL min^{-1} and the O_2 gas flow rate is 140 sccm. The total geometric area of the flow field in the cathode of our SE cell is 4 cm^2 .

Reviewer 2

This work presented an interesting point about the influence of alkali metal cations on the electrocatalytic H_2O_2 production in acidic media. The "shielding effect" induced by alkali metal cations is supposed to squeeze away the catalyst/electrolyte interfacial protons and thus prevent further reduction of generated H_2O_2 to water, thus achieving the high selectivity and stability. The authors performed various characterization techniques combined with calculations, and presented sufficient evidence to support their claims. Overall, this work is novel and can be published after minor revisions.

Response: We appreciate the reviewer's support of our work for publication, as well as the suggestions which have substantially improved the quality of our manuscript.

Comment 1: The pH value is adjusted to about 1 for the test. What would the catalytic performance be if the pH value further decreases or increases? For example, considering pH = 0 is used for the simulations, would the catalytic selectivity change significantly if using this pH value?

Response: We thank the reviewer for raising this critical point. In our simulation, we consider employed pH = 0 in our simulations, and found that the Na^+ shows "shield effect" at such low pH condition. To experimentally support it, we further increase the concentration of H_2SO_4 by using 1 M H_2SO_4 as electrolyte. As shown in Fig. S7 (also copied below), the H_2O_2 selectivity of carbon black BP 2000 catalyst at 1 M H_2SO_4 shows relatively good (~70%) selectivity under small current regions, but decreases dramatically at high current densities, reaching only ~15 % at $1,000 \text{ mA cm}^{-2}$. However, when we introduced 0.5 M Na_2SO_4 additive in 1 M H_2SO_4 as the electrolyte, the H_2O_2

selectivity was significantly improved, especially under high current densities. The Na^+ additive helped the carbon black catalyst to hold a high H_2O_2 selectivity plateau of $\sim 50\%$ until $1,000 \text{ mA cm}^{-2}$, suggesting a more than triple FE compared to that in pure $1 \text{ M H}_2\text{SO}_4$. The above observation is similar to what we observed using $0.1 \text{ M H}_2\text{SO}_4$ as electrolyte, indicating this is a general phenomenon by using cations to promote the H_2O_2 production in strong acidic conditions. The related descriptions have been included in the manuscript on Page 8, and Fig. S7 is included in the SI, which are also listed below for your reference.

Fig. S7. FE of H_2O_2 production through $2e^-$ -ORR in a flow cell by using $1 \text{ M H}_2\text{SO}_4 + 0.5 \text{ M Na}_2\text{SO}_4$ as catholyte. The H_2O_2 selectivity of carbon black BP 2000 catalyst at $1 \text{ M H}_2\text{SO}_4$ shows relatively good ($\sim 70\%$) selectivity under small current regions, but decreases dramatically at high current densities, reaching only $\sim 15\%$ at $1,000 \text{ mA cm}^{-2}$. However, when we introduced $0.5 \text{ M Na}_2\text{SO}_4$ additive in $1 \text{ M H}_2\text{SO}_4$ as the electrolyte, the H_2O_2 selectivity was significantly improved, especially under high current densities. The Na^+ additive helped the carbon black catalyst to hold a high H_2O_2 selectivity plateau of $\sim 50\%$ until $1,000 \text{ mA cm}^{-2}$, suggesting a more than triple FE compared to that in pure $1 \text{ M H}_2\text{SO}_4$. The above observation is similar to what we observed using $0.1 \text{ M H}_2\text{SO}_4$ as electrolyte, indicating this is a general phenomenon by using cations to promote the H_2O_2 production in strong acidic conditions.

At even lower pH electrolyte, i.e., $1 \text{ M H}_2\text{SO}_4$ solution (PH ~ 0), similar trends were also observed (Fig. S7), indicating the general phenomenon of cation promotion effect toward H_2O_2 production through ORR.

Comment 2: Would the H₂O₂ generation be always promoted if further more Na₂SO₄ is added?

Response: We appreciate the reviewer's important question. Yes, higher Na₂SO₄ concentration will further improve the H₂O₂ performance. As shown in Fig. 3b and 3c (copied below), the H₂O₂ selectivity gradually increases with increased Na⁺ concentrations. There is a trade-off relationship between the H₂O₂ selectivity-current density-cation concentration. In general, higher concentration of Na⁺ can drive larger 2e⁻-ORR currents without sacrificing the H₂O₂ selectivity, thus giving rise to a high H₂O₂ production rate. When we further increase the Na₂SO₄ concentration to 0.2 M, we found that the H₂O₂ FE can be maintained ~90% up to 600 mA cm⁻² (Fig. S3, listed below). Therefore, higher Na⁺ concentration will further improve the H₂O₂ performance.

Fig. 3. (b) The corresponding FEs and (c) production rates and partial current of H₂O₂ products under different cell voltages.

Fig. S3. FE of H₂O₂ production through 2e⁻-ORR in a flow cell using 0.1 M H₂SO₄ + 0.2 M Na₂SO₄ as catholyte.

Comment 3: The authors also explored other alkali metal ions in group IA and similar enhanced H₂O₂ selectivity was observed. What about the influence of other metal ions for the H₂O₂ generation, such as Mg²⁺ and Al³⁺?

Response: We appreciate the reviewer's important questions. Our molecular dynamic simulations suggest that the solvated alkali metal cations, compared to concentrated protons in acids, could preferentially be attracted to the catalyst/electrolyte interface and squeeze out local protons during reaction, suppressing the further reduction of as-generated H₂O₂ to H₂O. Our previous experiments also demonstrated that the alkaline metal ions (Li, Na, K, Cs) in IA group can greatly improve the H₂O₂ selectivity in acidic conditions. To further evaluate the influence of other cations on H₂O₂ production through ORR reaction, we conducted experiments by using 0.1 M H₂SO₄ + 0.01 M MgSO₄, 0.1 M H₂SO₄ + 0.01 M CaSO₄, 0.1 M H₂SO₄ + 0.01 M Al₂(SO₄)₃ as electrolyte. As shown in Fig. S9, after introducing a trace amount of Mg²⁺ additive (0.01 M MgSO₄) into the acidic electrolyte, the H₂O₂ selectivity was however significantly decreased, and eventually close to zero, which is greatly different from that of IA group cations (Li⁺, Na⁺, K⁺, Cs⁺). The differences are expected. Our previous study found that the IA group cations can help to regulate the local environment of the catalyst/electrolyte interface by repelling the local H⁺, giving rise to a local alkaline environment. However, at the alkaline environment, the Mg²⁺, Ca²⁺, and Al³⁺ could easily form solid Mg(OH)₂, Ca(OH)₂, and Al(OH)₃ precipitates, respectively, which could deposit on the catalyst surface and block the ORR process, inducing a low production activity and production rate. We also observed some white precipitations on the catalyst surface after our electrocatalytic test (Fig. S9 inset). The related figure is shown as Fig. S9 in the supplementary information and the related descriptions are included at Page 9 in the main manuscript, both are also shown below for your reference.

In addition, in our previous version, we used the "alkali metal cations" to indicate the IA group cations and distinguish them with other group cations. We also provide more descriptions of "alkali metal cations" in the revised manuscript.

Fig. S9. The FEs of H₂O₂ production through 2e⁻-ORR in a flow cell by using 0.1 M H₂SO₄ + 0.01 M MgSO₄, 0.1 M H₂SO₄ + 0.01 M CaSO₄, 0.1 M H₂SO₄ + 0.01 M Al₂(SO₄)₃ as catholyte, respectively. The inset picture is the carbon electrode after testing using 0.1 M H₂SO₄ + 0.01 M MgSO₄ as electrolyte. The white powders are deposited on the surface of the electrode.

Nevertheless, we find that the promotion effect is only limited to IA alkali metal cations (such as Li⁺, Na⁺, K⁺, Cs⁺), while the other cations (including the IIA cations such as Mg²⁺, Ca²⁺ and IIIA cation such as Al³⁺) decrease the H₂O₂ FE dramatically. This might due to the local environment change from acid to alkaline induced by the cation additives during ORR (will discuss the details in the simulation part). The alkaline local environment could induce the formation of solid metal hydroxide on the catalyst surface and block the ORR reaction, decreasing the H₂O₂ FE and production rate (Fig. S9).

Comment 4: Fig. 3b shows the enhanced performance in 0.1 M H₂SO₄ other than 0.1 M H₂SO₄ + 0.05M Na₂SO₄, which is contradictory with the description in the main text. The authors should carefully check this.

Response: We thank a lot for the reviewer's careful check. The figure legends were incorrectly used, which we modified in our revised version. Thanks again for the careful check. The revised version is also shown below:

Fig. 4b, The FEs and production rates of H_2O_2 in $0.1 \text{ M H}_2\text{SO}_4$ or $0.1 \text{ H}_2\text{SO}_4 + 0.05 \text{ M Na}_2\text{SO}_4$ electrolytes using CNT.

Comment 5: In the I-V curve, the authors used the j_{total} . It's recommended to provide a $J_{\text{H}_2\text{O}_2}$ -V curve for the better comparison.

Response: We appreciate the reviewer's important suggestion. We have revised our descriptions by providing both the production rate and partial current of H_2O_2 product. The production rate and partial current of H_2O_2 have a relationship with a coefficient, thus they share the same curves but different values and unit. We have modified Fig. 3 by providing both of them, which is also shown below:

Fig. 3. The effect of alkali metal cation concentration and species on electrosynthesis of H₂O₂ through 2e⁻-ORR. **a**, The I-V curve of ORR with different concentrations of cations (from 0 to 0.05 M Na₂SO₄) in a flow cell. **b**, The corresponding FEs and **(c) production rates and partial current of H₂O₂ products** under different cell voltages. With the increase of cation concentration, the H₂O₂ FE and production rate continuously increase at high current densities. **d**, The I-V curve of ORR in 0.1 M H₂SO₄ containing different types of cations (0.005 M X₂SO₄, X = Li⁺, Na⁺, K⁺, Cs⁺) in a flow electrolyte cell. **e**, The corresponding FEs and **(f) production rates and partial current of H₂O₂ products** under different cell voltages.

Comment 6: The authors could refer to some relevant work on the catalyst design for H₂O₂ generation. Such as Adv. Mater. 2021, 2104891.

Response: We appreciate the reviewer's suggestion. Some related works and valuable references have been cited in our revised manuscript.

Reviewer 3

The authors reported the addition of alkaline cations in electrolyte can significantly promote the production of H₂O₂. The promoted H₂O₂ FE over 80% and the stability over 5000 hours. The ab initio MD simulation examined the mechanism and found that cations can repel proton in electrolyte, shielding effect, to avoid the overoxidation reaction of H₂O₂ (forming H₂O); thus, improve the production of H₂O₂. Practical reactor for H₂O₂ generation is also demonstrated. The comments are listed below.

Response: We appreciate the reviewer's support of our work for publication, as well as the suggestions which have substantially improved the quality of our manuscript.

Comment 1: It is suggested that the authors can separate the scheme in Fig. 1a with other Fig. 1b-1d to smooth the description. Fig. 1a mentioned in Introduction, while other described in the next section.

Response: We appreciate the reviewer's important suggestion. We have now rearranged our scheme and figures accordingly based on the suggestion. Fig. 1a is now separated as Fig. 1, and Fig. 1b-1d is shown as Fig. 2. The corresponding descriptions have also been updated.

Comment 2: The authors should specifically emphasize the difference between present ORR forming H₂O₂ and another important ORR forming H₂O, which is cathodic reaction in the promising PEMFC and has been extensively studied. For example, the present ORR desires the formation of H₂O₂ and try to avoid the production of H₂O; however, the ORR in PEMFC application is completely opposite. Does that mean the "bad" results in the present study can be a "good" one in the ORR in PEMFC? The authors need to clarify the difference between those two ORR.

Response: We appreciate the reviewer's important suggestion. For ORR reaction, there are two reaction pathways, i.e., 2e⁻ and 4e⁻ pathways. The overall reaction equations in acidic conditions are shown below:

When pH > 11.6, the 2e⁻ pathway goes through the following reaction process:

The 2e⁻ and 4e⁻ pathways are competing with each other according to Equations 1~3. Therefore, shifting the reaction towards 2e⁻ pathway while suppressing the 4e⁻ process is one of the main strategies to improve the production efficiency of H₂O₂ through ORR. Nevertheless, this does not mean that the material not good for 2e⁻ is good for 4e⁻ reactions. For example, the commercial carbon black BP2000 catalyst shows poor 2e⁻ ORR performance in our work, and this catalyst is neither a good 4e⁻ ORR (acidic condition) catalyst in PEMFC, considering its poor activity.

Comment 3: The authors claimed that "higher concentration of Na⁺ can maintain larger 2e-ORR currents without sacrificing the H₂O₂ selectivity". Their results also showed that the highest concentration (0.1 M Na₂SO₄) shows the best H₂O₂ selectivity. Does that mean the selectivity can be further improved with even higher concentration (ex. 0.2 M)? Is there the optimized concentration to achieve the best selectivity (e.g. volcano plot)?

Response: We appreciate the reviewer's important suggestion. We found out that, in this manuscript, the Na⁺ can protect the catalytic interface to promote the H₂O₂ FE during the electrochemical ORR process. However, if the Na⁺ concentration is low, the FE will decrease when going to a high current density. For example, when using 5 mM Na⁺ as additive, the FE start to

decrease at 200 mA cm^{-2} . To achieve a better FE at a higher current, i.e., 400 mA cm^{-2} , more Na^+ ions are needed. For example, when we use 10 mM Na^+ as additive, the FE can reach to 80% at 400 mA cm^{-2} . The more Na^+ can make the 2e^- -ORR go to a high current density without sacrificing the FE. Based on the reviewer's suggestion, we also performed an experiment by using higher concentration of sodium ions ($0.1 \text{ M H}_2\text{SO}_4 + 0.2 \text{ M Na}_2\text{SO}_4$) as the electrolyte, and we found out that the FE can be maintained above 90% up to 500 mA cm^{-2} . Therefore, under high current density, more Na^+ will generate a high FE at high current density. In other words, higher Na^+ concentration will further improve the H_2O_2 performance. We do not believe there could be a volcano plot. The H_2O_2 FE by using $0.1 \text{ M H}_2\text{SO}_4 + 0.2 \text{ M Na}_2\text{SO}_4$ as electrolyte is included as Fig. S3 in supplemental information, and also listed below for your reference.

Fig. S3. FE of H_2O_2 production through 2e^- -ORR in a flow cell in $0.1 \text{ M H}_2\text{SO}_4 + 0.2 \text{ M Na}_2\text{SO}_4$ as catholyte.

Comment 4: The authors employed DFT to the mechanism study and found that the enhanced H_2O_2 selectivity corresponds to the repulsion between cation (Na^+) and proton (H^+) to ease the further protonation of H_2O_2 forming the unwanted H_2O . On the other hand, the experimental results found that other IA cations (Li^+ , K^+ and Cs^+) also showed similar result. There are some inconsistencies between those DFT and experimental results. According to DFT results, different sized cations should induce different repulsion to altered the H_2O_2 selectivity to some degree; however, the experiments observed the same promotion of all the cations (Fig. 2d-f). The authors can try with other cations with higher charges (e.g. Ca^{2+} , Mg^{2+} ...) or simulate with all the

experimental (1A) cations to clarify it. According to the experimental results, colligative property might be the more suitable mechanism in the present work.

Response: We appreciate the reviewer's great point here. First of all, there should not be any inconsistencies between our DFT results and our experimental results. Different sized cations will definitely induce different repulsion effects, which is also reflected from our experimental results. As shown in Fig. 3e (copied below), while all alkali cations can improve the H₂O₂ selectivity compared with the pure acidic electrolyte, their improvements are still slightly different. It is clear that Li⁺ presented the lowest H₂O₂ FEs among the other metal cations, and Cs⁺ presented the highest selectivity especially when the ORR current is high.

According to the reviewer's suggestion, we tried to replace two Na⁺ by one Mg²⁺, and ran the AIMD simulations similar to the case of Na⁺ (Fig. 5b). As shown in Fig. S27 (copied below), Mg²⁺ does not show the same effect of Na⁺. This is likely to be due to two reasons: 1) Mg²⁺ is more efficiently screened as the Mg²⁺ bonds stronger with O of H₂O than the case of Na⁺, which is evidenced by the significantly shorter distance between Mg and O (averaged distance $d_{\text{Mg-O}}=2.05$ Å vs. $d_{\text{Na-O}}=2.55$ Å); 2) the mole concentration of Mg²⁺ is only half of that of Na⁺ in the electrical double layer and the electrostatic repelling decays in the form of $1/r$, so there are more "screened" space for H⁺ in the electrical double layer. We note that this effect may essentially be a "colligative property" as the reviewer suggested. The related figure is included as Fig. S27 in the SI (also shown below), and the red-color descriptions here are included on Page 14 in the revised main manuscript.

Fig. 3 (e) The H₂O₂ FEs in 0.1 M H₂SO₄ containing different types of cations (0.005 M X₂SO₄, X = Li⁺, Na⁺, K⁺, Cs⁺) in a flow electrolyte cell.

Fig. S27. Final structure of the electrical double layer of water with Mg^{2+} and 2H^+ under $V_{\text{RHE}} = -1\text{V}$ after running AIMD for more than 4 picoseconds. The protons are colored cyan while the Mg is colored green.

Comment 5: The experimental results (Fig. 3) found that CNT, rGO and Zn-N-C are suitable catalysts for the present application. Also, their catalytic performance (production rate $\sim 6\text{ mmol/cm}^2/\text{h}$) are better than carbon black (BP2000) ($\sim 3\text{ mmol/cm}^2/\text{h}$). It is suggested the authors to use the better CNT, rGO or Zn-N-C to run to the latter tests of continuous production of H_2O_2 (Fig. 5) to achieve a better performance.

Response: We appreciate the reviewer's important suggestion. With the same concentration of Na^+ as additives, the performance of BP2000 and other catalysts are actually similar. For example, at 200 mA cm^{-2} , the carbon black BP2000 shows a production rate of $\sim 3\text{ mmol/cm}^2/\text{h}$. The catalytic performance of other carbon catalysts such as CNT, rGO, and Zn-N-C also show similar production rate of $\sim 3\text{ mmol/cm}^2/\text{h}$. When increasing the current density to 400 mA cm^{-2} , the production rate of BP is $\sim 6\text{ mmol/cm}^2/\text{h}$, which is also similar with that of other carbon catalysts. The BP 2000 is a commercial catalyst that has been more widely used in electrocatalysis.

REVIEWERS' COMMENTS

Reviewer #1 (Remarks to the Author):

The authors have cleared most of the questions raised in this revision. This work looks suitable for publication. I suggest the authors elaborate more on the statement below in their final version.

- Whether having Na⁺ ions in aqueous H₂O₂ is not a limitation in a real-world application
- Elaborate more about the activity and stability enhancement caused by cations in the main manuscript. Explanation in the point-by-point document is sound.

Some minor corrections needed are:

- P5L19 : Fig. 2b → 2c
- Captions read wrong in Figure S15,16

Reviewer #2 (Remarks to the Author):

All comments have been properly addressed. I recommend the acceptance of this manuscript in its current form.

Reviewer #3 (Remarks to the Author):

The manuscript has been revised accordingly. We have no further comments

Haotian Wang
Assistant Professor
Phone: 713-348-7221
Email: htwang@rice.edu

ChBE Department
Rice University
6100 Main Street
Houston, TX 77005

April 23, 2022

Manuscript Number: NCOMMS-21-50208A

Title: "Electrochemical oxygen reduction to hydrogen peroxide at practical rates in strong acidic media"

Authors: Xiao Zhang, Xunhua Zhao, Peng Zhu, Zachary Adler, Zhen-Yu Wu, Yuanyue Liu, Haotian Wang

Corresponding authors: Xiao Zhang, Yuanyue Liu and Haotian Wang

Response to reviewers' comments:

We thank the reviewers and editor for the constructive comments which have helped us to greatly improve our research and the quality of our manuscript. We have now included additional explanations to fully address the reviewer's concerns and suggestions. Below, we have addressed the points raised by reviewers.

Reviewer 1

The authors have cleared most of the questions raised in this revision. This work looks suitable for publication. I suggest the authors elaborate more on the statement below in their final version.

Response: We thank the reviewer's positive response and support our work for publication.

Comment 1

- Whether having Na^+ ions in aqueous H_2O_2 is not a limitation in a real-world application.

Response

We thank the reviewer's comment. In our study, we found only a trace amount of Na^+ additive (e.g., 5 mM Na_2SO_4) can greatly enhance the H_2O_2 FE and production rate. The required Na^+ concentration is too low, which has neglect effect towards the final H_2O_2 product for real-world applications, but a slightly increased production cost. In addition, for commercial H_2O_2 solutions, stabilizers are usually necessary to reduce the decomposition rate of H_2O_2 solutions. The Na^+ , in the forms of NaOOH , and NaOH , in

our design can also act as the stabilizers of aqueous H₂O₂.

Comment 2

- Elaborate more about the activity and stability enhancement caused by cations in the main manuscript. Explanation in the point-by-point document is sound.

Response

We appreciate the reviewer's important suggestion. We have included the following discussion in our revised manuscript (page 10).

We suppose the improved activity and stability are induced by the alkalization of local environment induced by the cation additives during ORR. During ORR process, the solvated alkali metal cations, compared to concentrated protons in acids, could preferentially be attracted to the catalyst/electrolyte interface, which gives rise to a local alkaline environment (will discuss the details in the simulation part). At alkaline conditions, the carbon-based catalyst typically shows higher activity and better stability compared to acidic conditions

Comment 2

Some minor corrections needed are:

- P5L19 : Fig. 2b → 2c

Response: We appreciate the reviewer's important suggestion. We have corrected it accordingly.

- Captions read wrong in Figure S15,16

Response: We appreciate the reviewer's valuable suggestion. We have corrected the captions of Fig. S15 and Fig. S16 accordingly.

Reviewer 2

All comments have been properly addressed. I recommend the acceptance of this manuscript in its current form.

Response: We thank the reviewer's positive response and support our work for publication.

Reviewer 3

The manuscript has been revised accordingly. We have no further comments.

Response: We thank the reviewer's positive response and support our work for publication.